# Structural plasticity of KIR2DL2 and KIR2DL3 enables altered docking geometries atop HLA-C

Shoeib Moradi[1,2,8], Sanda Stankovic[3,8], Geraldine M. O'Connor[4,8], Phillip Pymm[1,2], Bruce J. MacLachlan [1], Camilla Faoro [1], Christelle Retière[5,6], Lucy C. Sullivan[3], Philippa M. Saunders [3], Jacqueline Widjaja [3], Shea Cox-Livingstone[3], Jamie Rossjohn [1,2,7,9✉], Andrew G. Brooks [3,9✉] & Julian P. Vivian [1,2,9✉]

The closely related inhibitory killer-cell immunoglobulin-like receptors (KIR), KIR2DL2 and KIR2DL3, regulate the activation of natural killer cells (NK) by interacting with the human leukocyte antigen-C1 (HLA-C1) group of molecules. KIR2DL2, KIR2DL3 and HLA-C1 are highly polymorphic, with this variation being associated with differences in the onset and progression of some human diseases. However, the molecular bases underlying these associations remain unresolved. Here, we determined the crystal structures of KIR2DL2 and KIR2DL3 in complex with HLA-C*07:02 presenting a self-epitope. KIR2DL2 differed from KIR2DL3 in docking modality over HLA-C*07:02 that correlates with variabilty of recognition of HLA-C1 allotypes. Mutagenesis assays indicated differences in the mechanism of HLA-C1 allotype recognition by KIR2DL2 and KIR2DL3. Similarly, HLA-C1 allotypes differed markedly in their capacity to inhibit activation of primary NK cells. These functional differences derive, in part, from KIR2DS2 suggesting KIR2DL2 and KIR2DL3 binding geometries combine with other factors to distinguish HLA-C1 functional recognition.

[1] Infection and Immunity Program and Department of Biochemistry and Molecular Biology, Biomedicine Discovery Institute, Monash University, Clayton, VIC, Australia. [2] Australian Research Council Centre of Excellence in Advanced Molecular Imaging, Monash University, Clayton, VIC, Australia. [3] Department of Microbiology and Immunology, Peter Doherty Institute for Infection and Immunity, The University of Melbourne, Parkville, VIC, Australia. [4] School of Medicine, University of Central Lancashire, Preston, UK. [5] Etablissement Français du Sang, Nantes, Nantes, France. [6] Université de Nantes, INSERM U1232 CNRS, CRCINA, Nantes, France. [7] Institute of Infection and Immunity, Cardiff University, School of Medicine, Heath Park, Cardiff, UK. [8] These authors contributed equally: Shoeib Moradi, Sanda Stankovic, Geraldine M. O'Connor. [9] These authors jointly supervised this work: Jamie Rossjohn, Andrew G. Brooks, Julian P. Vivian. ✉email: jamie.rossjohn@monash.edu; agbrooks@unimelb.edu.au; julian.vivian@monash.edu

Natural killer (NK) cells discriminate between healthy and infected or transformed cells using an array of germline-encoded inhibitory and activating receptors[1,2]. Central to this process is the interaction between the killer cell immunoglobulin-like receptors (KIR) and their cognate ligands, human leukocyte antigen class I molecules (HLA-I)[3–5]. In humans, three highly polymorphic *loci* encode classical *HLA-I* genes, *HLA-A*, *-B* and *-C*, the latter being particularly important for regulating NK cell function[6]. *HLA-C* alleles are divided into two groups, *C1* and *C2* based on dimorphisms at positions 77 and 80 located on the α1-helix of the antigen-binding cleft with the dimorphism at position 80 in particular being a critical specificity determinant for some KIR. HLA-C1 allotypes, which are characterised by an asparagine at position 80 are the preferred ligands for the inhibitory receptors KIR2DL2 and KIR2DL3 whereas HLA-C2 allotypes encode a lysine at this position and are recognised by KIR2DL1[7,8]. A primary specificity difference between KIR2DL1, KIR2DL2 and KIR2DL3 has been attributed to the presence of a methionine at position 44 in KIR2DL1 compared with a lysine in KIR2DL2 and KIR2DL3[8].

Analyses of the specificity of KIR2DL1, -2DL2 and -2DL3 has been assessed using KIR-Fc-fusion proteins[9–12] and suggested that KIR2DL2 and KIR2DL3 are more broadly HLA-C1/C2 cross-reactive than KIR2DL1[10,11]. Such studies have also suggested that KIR2DL2 has a greater avidity than KIR2DL3 for *HLA-C1* group allotypes, a greater breadth of binding across the *HLA-C2* group along with the capacity to interact with a broader array of peptides[9,12,13]. While the KIR2DL2/3 receptors share ~94% sequence identity and segregate as alleles of a single locus, they are inherited on different KIR haplotypes[14–18], which has implications for NK cell function. For instance, the expression of KIR2DL1 is epistatically suppressed by the expression of KIR2DL2, a phenomenon not observed with KIR2DL3[19]. Similarly, *KIR2DS2* is in linkage disequilibrium with *KIR2DL2* and not *KIR2DL3*[20]. Moreover, these differences are associated with different clinical outcomes. For example, in the presence of *HLA-C1* alleles, *KIR2DL3* but not *KIR2DL2* has been associated with clearance of hepatitis-C infection[21,22] as well as the progression of ulcerative colitis[23]. However, the mechanistic basis of these selective associations is not well defined.

The structures of KIR2DL1 in complex with peptide (p)-HLA-C*04:01 and KIR2DL2 bound to pHLA-C*03:04 show a conserved binding mode, with the C-terminus of the peptide sitting at the junction of the two domains of the KIR (D1 and D2) and the D1 and D2 domains of the KIR2DL receptors docking on the α1- and α2-helices of the HLA-C, respectively. While the KIR2DL contacts with the α2-helix are conserved, interactions with both the peptide and the α1-helix vary, consistent with the ability of KIR2DL1 and 2DL2 to discriminate between HLA-C1 and HLA-C2 allotypes and different peptides bound to individual HLA-I allotypes[9,12,24–28].

Presently, a molecular basis for any functional differences between KIR2DL2 and KIR2DL3 is unknown. Here, we determined the crystal structures of KIR2DL2 and KIR2DL3 in complex with pHLA-C*07:02, which showed that KIR2DL2 and KIR2DL3 engage HLA-C1 with different docking angles. Moreover, via mutagenesis and binding studies, we show that KIR2DL2 and KIR2DL3 utilise similar frameworks of contact residues to recognise HLA-C, yet do so in a manner that is dependent on the allotype of HLA-C1 molecule itself. These structural distinctions provide insights into how KIR2DL2 and KIR2DL3 can discriminate between similar HLA-C allotypes. Nevertheless, in vitro analyses of primary NK cells bearing either KIR2DL2 or KIR2DL3 suggest that the functional impact is subtle and that understanding differences between KIR2DL2 and

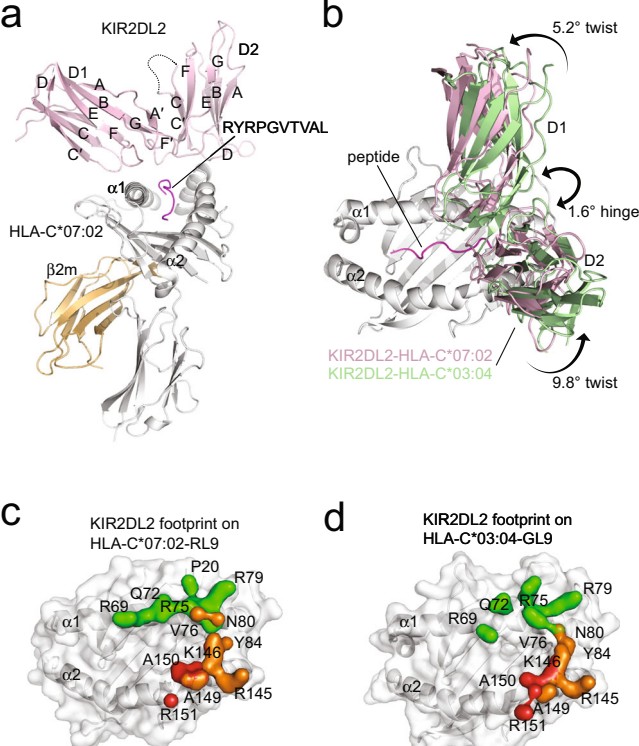

**Fig. 1 Structural comparison of KIR2DL2 in complex with HLA-C*07:02-RL9 and HLA-C*03:04-GL9. a** Overall structure of KIR2DL2 in complex with HLA-C*07:02-RL9. The complex is represented in cartoon format with KIR2DL2 coloured pink, HLA-C*07:02 coloured grey, β2 m coloured orange and the peptide coloured magenta. **b** Comparison of KIR2DL2 in complex with HLA-C*03:04-GL9 (KIR coloured green) and HLA-C*07:02-RL9 (KIR coloured pink). The relative difference in hinge and twist angles of the KIR is shown. Footprints of KIR2DL2 on HLA-C*07:02-RL9 (**c**) and HLA-C*03:04-GL9 (**d**) coloured by domain binding. D1 contacts coloured green, D1-D2 interdomain loop contacts coloured red and D2 contacts coloured orange.

KIR2DL3 in clinical settings may be confounded by co-expression of activating receptors with specificity for C1 allotypes.

## Results

**KIR2DL2 shows distinct binding to HLA-C*03:04 and HLA-C*07:02.** We determined the crystal structure of KIR2DL2 in complex with HLA-C*07:02 presenting a self-peptide derived from histone H3 (RL9, residues 40–48, RYRPGTVAL)[29]. To date, crystal structures of KIR2DL1, KIR2DL2, KIR2DS2 with single HLA allotypes as well as KIR3DL1 in complex HLA-B*57:01 and -B*57:03 have been determined[24,25,30,31]. The structure of KIR2DL2 in complex with HLA-C*03:04-GAVDPLLAL (GL9) was previously determined[24], and together with our structure of KIR2DL2 in complex with HLA-C*07:02-RL9 provided an opportunity to compare how allelically related KIR2D recognise distinct HLA-I allotypes (Figs. 1a, b and 2a, b).

The KIR2DL2-HLA-C*07:02-RL9 ternary complex was determined to 3.1 Å resolution and displayed unambiguous density at the KIR2DL2/HLA-C interface allowing for ready interpretation (Supplementary Table 1, Supplementary Fig. 1). The crystal lattice contained two ternary complexes in the asymmetric unit that were highly similar (root mean square deviation (r.m.s.d.) <0.4 Å over all Cα positions), and accordingly, one copy of the complex was used for analysis. Overall, the binding of KIR2DL2 to HLA-C*07:02-RL9 was reminiscent of previously determined

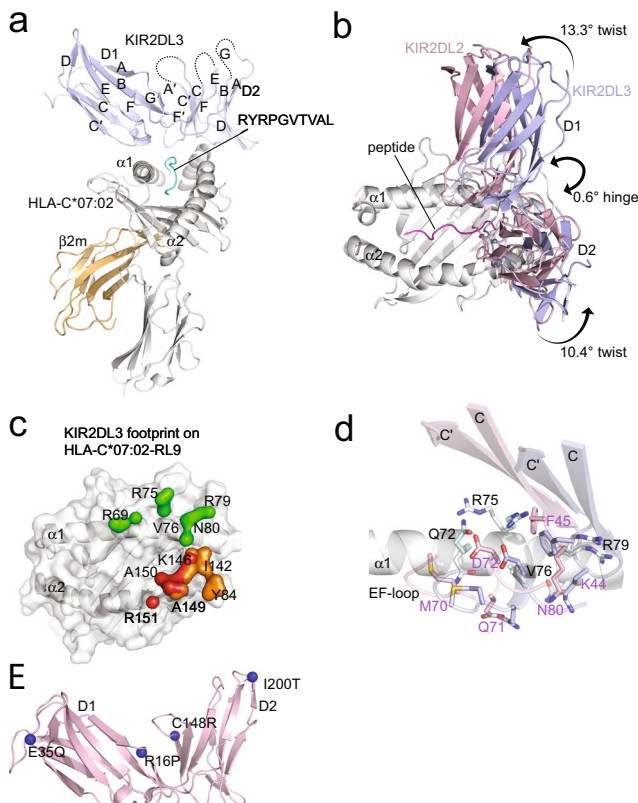

**Fig. 2 Structural comparisons of KIR2DL2 and 2DL3 in complex with HLA-C\*07:02-RL9. a** Overall structure of KIR2DL3 in complex with HLA-C\*07:02-RL9. The complex is represented in cartoon format with KIR2DL2 coloured blue, HLA-C\*07:02 coloured grey, β2m coloured orange and the peptide coloured cyan. **b** Comparison of KIR2DL2 (pink) and 2DL3 (blue) in complex with HLA-C\*07:02-RL9. The relative difference in hinge and twist angles of the KIR is shown. **c** Footprint of KIR2DL3 on HLA-C\*07:02-RL9 coloured by domain binding. D1 contacts coloured green, D1-D2 interdomain loop contacts coloured red and D2 contacts coloured orange. **d** Comparison of KIR2DL2 (pink) and KIR2DL3 (blue) D1 contacts to the α1-helix of HLA-C\*07:02-RL9 (KIR2DL2 contacts coloured grey, KIR2DL3 contacts coloured cyan). The ~3 Å shifts of the D1 C-C′ and E-F loops are highlighted. **e** Mapping of sequence differences to the structures of KIR2DL2/3. Differences highlighted as blue spheres.

KIR2D-pHLA ternary complexes, with the elbow-angle of the receptor formed by the junction of the KIR2D C-type immunoglobulin domains (D1 and D2) characteristically straddling the centre of the HLA peptide-binding groove and positioned over the C-terminus of the peptide (Fig. 1a). This arrangement placed the KIR D1 domain over the HLA α1-helix that was bound via three KIR2DL2 loops, A-B, C-C′ and E-F, with canonical contacts to the HLA-lineage defining Asn80 by the C-C′ residue Lys44 (Fig. 1a, c, Supplementary Fig. 2a, Supplementary Table 2). The HLA-I α2-helix was recognised via two loops of the D2 domain; the B-C loop spanning Ser132 to Asp135 and the F-F′ loop that spans Phe181 to Glu187 as well as the D1-D2 interdomain loop (Fig. 1a, c, Supplementary Fig. 2b, Supplementary Table 2).

Comparing the structure of KIR2DL2-HLA-C\*07:02-RL9 with the previously determined structure of KIR2DL2-HLA-C\*03:04-GL9 (PDB accession code 1EFX[24]) revealed very similar docking modes. The buried surface area (B.S.A.) of the two ternary complexes were similar (KIR2DL2-HLA-C\*03:04 ~1600 Å², KIR2DL2-HLA-C\*07:02 ~1700 Å²). All residues on KIR2DL2 that contacted HLA were also conserved between the KIR2DL2 complexes. Indeed, of the 21 polymorphisms that distinguish HLA-C\*03:04 from HLA-C\*07:02, nine map to the peptide-binding groove and while impacting peptide repertoire, none directly contacted KIR2DL2. Moreover, there were conserved contacts to the P7 and P8 positions of the GL9 and RL9 peptides, mediated by Gln71[KIR2DL2] and Leu104[KIR2DL2] (Supplementary Fig. 3). Notwithstanding the sequence differences, HLA-C\*03:04 and C\*07:02 superposed very closely with a r.m.s.d. of 0.9 Å (over Cα residues 1–180). Yet, there was a difference observed at the binding site for the D2 domain of KIR2DL2 on the α2-helix linker (residues 149–153) that deviated by 1.8 Å (Supplementary Fig. 2c).

While there was a highly similar hinge angle for the KIR2DL2 receptor (76.1° for 2DL2-HLA-C\*07:02 and 77.7° for 2DL2-HLA-C\*03:04) (Fig. 1b), the docking of KIR2DL2 differed about the twist of the receptor by 5.2° for D1 domains and 9.8° for D2 domains (Fig. 1b, Supplementary Fig. 4), where twist is defined as the rotation of the KIR2DL domains about their short axis, perpendicular to the short axis that defines the hinge angle (shown in Fig. 1b). As a result, this led to some subtle differences in the contact points between these two HLA-C allomorphs (Fig. 1a–c, Supplementary Fig. 2a, b, Supplementary Table 2). Namely, KIR2DL2 made more unique contacts to HLA-C\*07:02. Specifically, Pro20[C\*07:02] interacted with Phe45[KIR2DL2], Val76[C\*07:02] interacted with and Lys44[KIR2DL2] and Glu187[KIR2DL2] and Lys146[C\*07:02] and Ala149[C\*07:02] contacted Glu187[KIR2DL2] and Tyr134[KIR2DL2], respectively (Fig. 1c, d, Supplementary Fig. 2a). By contrast, the only contacts observed in the KIR2DL2-HLA-C\*03:04 ternary complex and not in the C\*07:02-RL9 were, Arg75[C\*03:04] contacted Asp72[KIR2DL2], and Tyr105[KIR2DL2] contacted P7-Val of the RL9 peptide (Fig. 1c, d, Supplementary Figs. 2b and 3). Taken together, KIR2DL2 used the same framework of D1 and D2 residues to recognise subtly different regions on these distinct HLA-C allotypes, with the twist angle providing the basis of the differences between these two ternary KIR2DL2-pHLA-C complexes.

**KIR2DL2 and KIR2DL3 show distinct docking modes to pHLA-C\*07:02.** Next, we next determined the crystal structure of KIR2DL3\*001 in complex with HLA-C\*07:02 bound to the RL9 peptide. This allowed for structural comparison of two KIR2DL allomorphs bound to the same HLA-peptide complex. The KIR2DL3 ternary complex was determined to 2.5 Å resolution (Fig. 2a, Supplementary Table 1).

Overall, the binding of KIR2DL3 to HLA-C\*07:02-RL9 largely recapitulated that observed for KIR2DL2-HLA-C\*07:02-RL9 (and other KIR2D-HLA complexes)[24,25,30]. Namely, the D1 domain interacted with the HLA α1-helix via the A-B, C-C′ and E-F loops, whilst two recognition loops of the D2 domain (B-C and F-F′) and the D1-D2 interdomain loop made contacts with the HLA α2-helix (Fig. 2a, Supplementary Table 2). KIR2DL2 and KIR2DL3 adopted highly similar hinge angles of 77.7° and 77.1°, respectively (Fig. 2b). Individually, the D1 and D2 domains aligned with r.m.s.d. values of 0.45 Å (residues 1–102) and 0.61 Å (residues 108–200), respectively. Yet, despite this structural homology, differences were observed in the relative juxtapositioning of the D1 and D2 domains of KIR2DL2/3. Specifically, the structural differences between KIR2DL2 and KIR2DL3 centred on the twist of the D1 and D2 domains that differed by 13.3° for the D1 and 10.4° for the D2 domains (Fig. 2b, c).

The extracellular domains of KIR2DL2\*001 and KIR2DL3\*001 differ at positions 16, 35, 148 and 200 (Fig. 2e). Of these, positions 35 (Glu35Gln) and 200 (Ile200Thr) are located distal to the hinge on the membrane-proximal side of the D1 and D2 domains, respectively (Fig. 2e). In contrast, both Arg16Pro (D1 A-A′ loop) and Cys148Arg (D2 C-C′ loop) are positioned proximal to the

D1-D2 hinge with these loops spanning the D1-D2 domain junction. Here, Arg16$^{KIR2DL2}$ reached across the D1-D2 interface adjacent the D2 C-C′ loop binding His146$^{KIR2DL2}$, while Pro16$^{KIR2DL3}$ made no direct interdomain interaction (Supplementary Fig. 5a). Notably, KIR2DL1*001 (PDB code 1IM9[25]) contains a Pro at residue 16, and possesses an Arg at position 148. Given that KIR2DL1 displays a highly similar twist angle to KIR2DL3, with relative differences of 3.6° on the D1 and 0.8° for the D2 domain (Supplementary Fig. 4c, d), this suggests that the polymorphisms at positions 16 and 148 are likely drivers of the relative positioning of the D1-D2 domains.

This differing D1-D2 domain configuration resulted in the centre of mass of the D1 domain of KIR2DL3 sitting ~3 Å more towards the C-terminus of the peptide-binding groove of HLA-C*07:02 relative to that observed for KIR2DL2. It also resulted in the same framework of KIR2DL3 residues binding subtly different regions of HLA-C*07:02-RL9. Namely, for the D1 domain, KIR2DL2 made more extensive contacts to the HLA α1-helix (20% more B.S.A. Fig. 2c). These additional contacts comprised residues Lys44$^{KIR2DL2}$ and Phe45$^{KIR2DL2}$ (C-C′ loop) that interacted with Val76$^{C07:02}$ and Pro20$^{C07:02}$, respectively (Fig. 2d). Furthermore, the D1 E-F loop shifted ~3.4 Å between the complexes (Fig. 2b, d), which enabled additional contacts involving Lys44$^{KIR2DL2}$, Met70$^{KIR2DL2}$, Gln71$^{KIR2DL2}$ and Asp72$^{KIR2DL2}$ contacting Gln72$^{C07:02}$ and Val76$^{C07:02}$ (Fig. 2d, Supplementary Table 2). For KIR2DL3, there was an additional contact between Asp72$^{KIR2DL3}$ (E-F loop) and Arg75$^{C07:02}$ (Fig. 2d). Similarly, the D1-D2 interdomain loop and the D2 domains of KIR2DL2 and KIR2DL3 also exhibited differences in their interactions with the α2-helix (Figs. 1c and 2c, Supplementary Fig. 5b). For the D1-D2 interdomain loop, a difference was observed at Glu106$^{KIR2DL2}$ interacting with Arg151$^{C07:02}$ (Figs. 1c and 2c, Supplementary Fig. 5b). Within the D2 domain, Ser184$^{KIR2DL2}$, Glu187$^{KIR2DL2}$ and Tyr134$^{KIR2DL2}$ made unique contacts to Lys146$^{C07:02}$ and Ala149$^{C07:02}$ (Fig. 2c, Supplementary Fig. 5b). For KIR2DL3, unique contacts were formed between Phe181$^{KIR2DL3}$ and Asp183$^{KIR2DL3}$ and Arg145$^{C07:02}$ and Ile142$^{C07:02}$, respectively (Figs. 1c and 2c, Supplementary Fig. 5b, Supplementary Table 2). Regarding peptide contacts, KIR2DL2 and KIR2DL3 formed conserved peptide contacts to the P7 and P8 positions of the RL9 peptide via resides Leu104$^{KIR2DL2/2DL3}$ and Gln71$^{KIR2DL2/2DL3}$ respectively, yet KIR2DL2 made an additional contact to P7-Leu via Leu104$^{KIR2DL2}$ (Supplementary Table 2, Supplementary Fig. 2). Accordingly, KIR2DL2 and KIR2DL3 use a highly conserved constellation of residues to interact with HLA-C molecules, yet the differing juxtapositioning of the D1-D2 domains provides a framework from which might impact recognition of different HLA-C1 allotypes.

**HLA-C*03:04 and HLA-C*07:02 are not equivalent for KIR2DL recognition.** To evaluate the extent to which structural variations across KIRDL2/3-HLA-C interactions correlated with binding affinity, we probed the interaction of KIR2DL2 and KIR2LD3 to HLA-C*03:04-GL9 and HLA-C*07:02-RL9. HLA-C tetramers were used to stain 293T cells expressing matched levels of either FLAG-tagged KIR2DL2/3 or cells bearing one of 14 alanine point mutations at positions implicated in ligand recognition and the proportion of tetramer-positive cells normalised against the proportion of cells that stained with FLAG-specific mAb (Fig. 3a–c). A comparison of the binding of the HLA-C*03:04-GL9 tetramer to KIR2DL2 and KIR2DL3 mutants revealed highly similar patterns of contact residue dependence (Fig. 3b). Specifically, mutation of residues Phe45, Leu104, Tyr105, Asp135, Phe181 and Asp183 all significantly impacted binding of the HLA-C*03:04 tetramer to both KIR2DL2 and KIR2DL3 (Fig. 3b). Two

substitutions resulted in a statistically significant difference in binding between KIR2DL2 and KIR2DL3 to HLA-C*03:04, namely Lys44Ala ($P = 0.0451$) and Ser133Ala ($P = 0.0096$).

Similarly, staining of cells expressing wild-type or mutant KIR2DL2 and KIR2DL3 with HLA-C*07:02-RL9 tetramer also showed highly similar patterns of contact residue dependence between KIR2DL2 and KIR2DL3. Nevertheless, the pattern of dependence differed significantly from that observed for HLA-C*03:04-GL9 with the staining of HLA-C*07:02 being less perturbed by mutation (Fig. 3c). Specifically, alanine substitutions at Phe45, Leu104, Ser133 and Asp183, all of which abrogated HLA-C*03:04 recognition, were tolerated by HLA-C*07:02 (Fig. 3b, c). Indeed, the Tyr105Ala and Phe181Ala mutations were the only substitutions that abrogated binding to the HLA-C*07:02 tetramer (Fig. 3c). In addition, in contrast to HLA-C*03:04 tetramer binding, for the HLA-C*07:02-RL9 tetramer, only the Asp135Ala mutation showed a statistically significant difference in binding to KIR2DL2 and KIR2DL3, the latter being less tolerant of this substitution. Collectively, the data revealed clear differences in the recognition mode of KIR2DL2 and KIR2DL3 for HLA-C*03:04-GL9 and HLA-C*07:02-RL9. Moreover, the mutational analyses highlighted a degree of plasticity in KIR2D recognition of C1 allotypes, where polymorphisms in the KIR can alter the basis of ligand binding that may ultimately impact the range of C1/peptide combinations that can be recognised.

**Distinct general characteristics of HLA-C recognition by KIR2DL2 and KIR2DL3.** As the above observations were based on the KIR2DL recognition of HLA-C1 allotypes bound to a single peptide, we next examined the extent to which differences between KIR2DL2 and KIR2DL3 could be modulated by peptide repertoire. We measured KIR2DL2 and KIR2DL3 binding via SPR to a panel of twelve HLA-C*07:02-RL9 peptide substitutions which spanned a variety of amino acids in the key KIR2DL contact regions P7 and P8 (Table 1 and Supplementary Fig. 6). With the exception of P8F and P8V, each substitution resulted in reduced affinity binding to both KIR2DL2 and KIR2DL3 (Table 1). For example, inclusion of the acidic resiude Glu at P7 or P8 was detrimental to binding of both KIR2DLs in line with previous findings[32]. However, both KIR2DL2 and KIR2DL3 shared similar binding preferences across the peptides in this panel (Table 1). Thus, at least in the context of the HLA-C*07:02-RL9 peptide, there were no marked differences between KIR2DL2 and 2DL3 with respect to their sensitivity to P7 and P8 substitutions.

To evaluate KIR2DL binding in a broader context, soluble tetrameric forms of KIR2DL2 and KIR2DL3 molecules were generated and assessed for their ability to bind a panel of 97 bead-bound HLA allotypes including 18 HLA-C1/C2 molecules, each presenting a broad repertoire of peptides. As expected, both KIR2DL2 and KIR2DL3 preferentially bound HLA-C1 molecules over HLA-C2 molecules and exhibited minimal reactivity with HLA-A and -B allotypes in the panel (Fig. 4a–c). Nevertheless, there was considerable variation in the hierarchy of avidity for different C1 allotypes. For example, KIR2DL2 bound HLA-C*14:02 with only 14.7% of the avidity observed for the top binding C1 allomorph, HLA-B*73:01. Indeed, C2 group allomorphs, such as HLA-C*15:02 and C*17:01, bound KIR2DL2 at ~55% of the top avidity, higher than a number of C1 group allomorphs (e.g. HLA-C*12:03 and HLA-C*14:02) consistent with previous observations suggesting KIR2DL2 had the potential to cross react with C2 allotypes (Fig. 4a, b).

KIR2DL3 also preferentially bound HLA-C1 molecules over HLA-C2 allotypes with broad recognition across HLA-C1 allomorphs. Namely, of the 11 HLA-C1 allotypes examined,

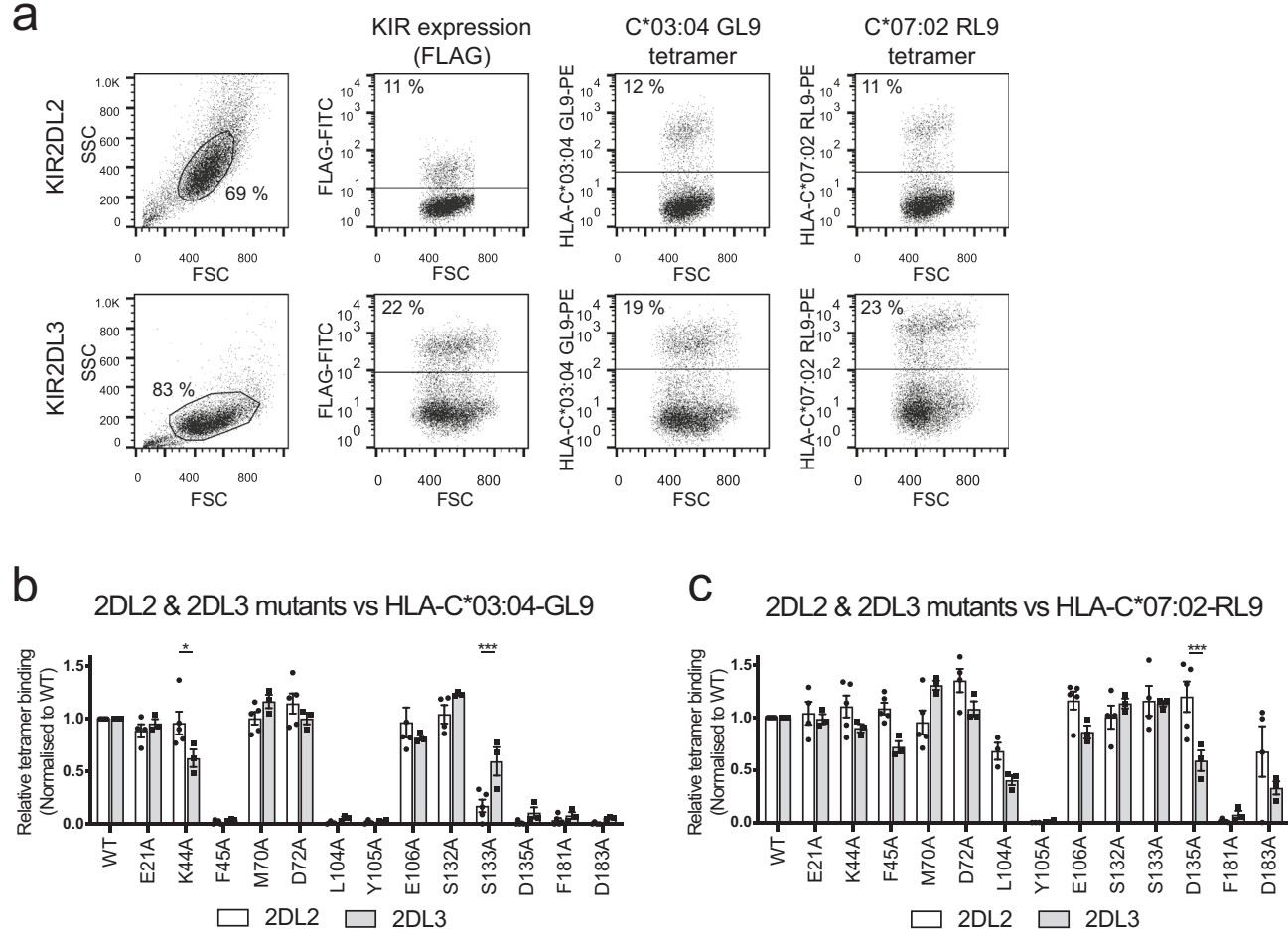

**Fig. 3 Mapping the interaction between KIR2DL2/2DL3 to HLA-C*03:04-GL9 and C*07:02-RL9.** 293T cells were transfected with plasmids encoding either wild-type or mutant FLAG-tagged KIR2DL2 and KIR2DL3 and then stained with a FLAG-specific mAb or HLA-C*03:04-GL9 and HLA-C*07:02-RL9 tetramers. Representative dot plots showing staining to cells expressing wild-type KIR2DL2 and KIR2LD3 (**a**). Relative binding of wild-type (WT) KIR2DL2 and KIR2DL3 transfected 293T cells and their respective alanine mutants to phycoerythrin-conjugated tetramers of HLA-C*03:04-GL9 (**b**) and HLA-C*07:02-RL9 (**c**). Transfected cells were stained for FLAG as a marker of KIR2DL expression and separately stained with HLA tetramers. Percentage tetramer+ was normalised to percentage FLAG + to account for differing KIR2DL transfection efficiency between mutants. The data represent a minimum of $n = 3$ independent experiments; bar line = mean, error bars = standard error of the mean (SEM); individual data points = circles (2DL2) or squares (2DL3). For each independent replicate, relative tetramer staining was normalised to staining of WT KIR2DL. Statistically significant differences in ligand recognition between corresponding mutants of KIR2DL2 and KIR2DL3 as measured by Two-way ANOVA Šidák's multiple comparison test are highlighted. 2DL2 ($n = 5$) vs. 2DL3 ($n = 3$) binding to C*03:04-GL9 K44A: *$P = 0.0127$; 2DL2 ($n = 5$) vs. 2DL3 ($n = 3$) binding to C*03:04-GL9 S133A: ***$P = 0.0005$; 2DL2 ($n = 5$) vs. 2DL3 ($n = 3$) binding to C*07-02-GL9 D135A: ***$P = 0.0007$.

KIR2DL3 bound 9 with greater than 75% of the maximum avidity, with only HLA-C*14:02 interacting at modest levels (43% of the maximum avidity). Moreover, in the case of KIR2DL3, there was a clearer distinction between specificity for C1 and C2 allotypes with only the HLA-C*14:02 allotype interacting more weakly than the highest binding C2 allotypes (Figs. 4a, c).

Comparing the binding preferences between KIR2DL2 and KIR2DL3 at the level of individual HLA-C allomorphs showed considerable variation. For example, the majority of the HLA-C1 allomorphs tended to be recognised with higher avidity by KIR2DL3, with an average relative avidity across the group of 82.2% compared with 68.3% for KIR2DL2. Moreover, individual HLA-C allomorphs also showed differences in binding preference between the two KIR2DL receptors. For example, HLA-C*08:01 and HLA-C*12:03, bound considerably better to KIR2DL3 than to KIR2DL2, whilst HLA-C*07:02 was bound with higher avidity by KIR2DL2 (Fig. 4a–c). Accordingly, the data show that there were distinct differences in the capacity of these allelically distinct KIR2D to recognise individual C1 allotypes. Thus polymorphism

across both the *KIR* and *HLA* loci resulted in differences in the strength of these receptor/ligand interactions.

**Allotypic differences in HLA-C recognition by NK cells expressing KIR2DL2 and KIR2DL3.** To understand how differences in KIR2D/HLA-C interactions potentially translated to functional outcomes, we initially compared NK cell recognition of three cell lines isolated from Burkitt lymphoma patients; Raji (HLA-C*03:04/C*04:01-C1/C2), Ramos (HLA-C*16:01-C1) and Namalwa (HLA-C*07:01/07:02-C1/C1). Flow cytometric analyses showed that Raji and Ramos expressed the similar levels of HLA-C with Namalwa having slightly lower levels of expression as assessed by staining with the DT9 mAb (Fig. 5a). Primary NK cells were co-cultured with the lymphoma lines or the HLA-I-deficient cell line 721.221 (221) and then stained with an antibody panel that could distinguish between three populations of KIR2D-expressing cells; KIR2DL1/S1, KIR2DL1⁻KIR2DL2/S2+KIR2DL3⁻(KIR2DL2/S2+) or KIR2DL1⁻KIR2DL2⁻KIR2DL3+ (KIR2DL3+) (Fig. 5b and Supplementary Fig. 7). Coculture of primary

NK cells with Raji cells resulted in relatively similar levels of activation of cells expressing KIR2DL1/S1, KIR2LD2/S2 or KIR2DL3 consistent with their expression of both C1 and C2 allotypes (Fig. 5c, d). In

**Table 1 KIR2DL2 and KIR2DL3 binding to HLA-C*07:02-RL9-P7/P8 mutant peptide panel.**

| Peptide | KIR2DL2 | | KIR2DL3 | | | |
| --- | --- | --- | --- | --- | --- | --- |
| | $K_D$ (μM) | SD (±μM) | $K_D$ (μM) | SD (±μM) | p value | q value |
| WT | 4.9 | 0.3 | 4.2 | 0.3 | 0.31 | 0.39 |
| P7E | n.d. | n.d. | n.d. | n.d. | n.d. | n.d. |
| P7F | 14.4 | 0.5 | 12.4 | 1.6 | 0.23 | 0.37 |
| P7G | 27.9 | 2.1 | 21.0 | 1.6 | 0.07 | 0.27 |
| P7A | 27.7 | 3.3 | 20.6 | 0.7 | 0.10 | 0.27 |
| P7N | 29.7 | 5.7 | 21.5 | 1.5 | 0.19 | 0.35 |
| P7R | 17.6 | 0.9 | 14.4 | 3.0 | 0.28 | 0.39 |
| P8E | n.d. | n.d. | n.d. | n.d. | n.d. | n.d. |
| P8F | 6.5 | 0.4 | 5.1 | 0.4 | 0.07 | 0.27 |
| P8G | 18.9 | 0.7 | 11.7 | 0.8 | 0.01 | 0.12 |
| P8V | 7.5 | 1.0 | 5.6 | 0.6 | 0.15 | 0.33 |
| P8N | 20.0 | 5.8 | 19.9 | 2.0 | 0.99 | 0.99 |
| P8R | 23.9 | 1.5 | 22.8 | 1.5 | 0.54 | 0.60 |

WT wild-type RL9 peptide (RYRPGTVAL), n.d. not determined, KD equilibrium dissociation constant, SD standard deviation of calculated KD value.
KD values calculated via steady-state analysis of n = 2 injections of serially diluted KIR2DL analyte described in full in Supplementary Fig. 6. Representative values of n = 2 independent experiments are shown. Differences between KIR2DL2 and KIR2DL3 binding to each ligand was assessed by multiple unpaired two-sample t-test using a false discovery rate (FDR) approach (two-stage step-up method). p and q values for each comparison are shown. With a desired FDR (Q) of 1%, no discoveries were observed.

contrast, both the Ramos and Namalwa cell lines, which lack C2 allotypes, stimulated robust levels of degranulation of cells that expressed KIR2DL1/S1. Despite modest differences in the level of HLA-C expression, the extent of inhibition of KIR2DL2/S2 expressing cells as reflected in the proportion of cells that expressed CD107a compared to stimulation with 721.221 cells was similar between Ramos and Namalwa. Similarly, the responses of cells that expressed KIR2DL3 were also comparable following coculture with Ramos and Namalwa. Interestingly, the degranulation response of the KIR2DL2/S2⁺ population to both Ramos and Namalwa was larger than that observed for cells expressing KIR2DL3.

To extend the range of HLA-C allotypes assessed, NK cells from donors who were heterozygous for KIR2DL2/3 were assayed for their capacity to recognise .221 cells transfected with seven distinct HLA-C allotypes. While .221 cells induced a robust response from all three NK cell populations, expression of the C2 ligand HLA-C*06:02, impaired activation of NK cells expressing KIR2DL1 but had a less marked impact on the degranulation responses of cells expressing either KIR2DL2/S2 or -2DL3 (Fig. 5e, f). In contrast, as expected, expression of HLA-C1 allotypes inhibited the responses of cells expressing KIR2DL2/S2 and KIR2DL3 but had limited impact on the activation of cells expressing KIR2DL1/S1. As observed in the bead-based binding assays, there were substantial differences in the capacity of individual HLA-C1 allotypes to inhibit activation of KIR2DL2/2DL3⁺ cells. For example, the expression of HLA-C*03:02, HLA-C*07:02 or HLA-C*16:01 strongly inhibited degranulation of cells expressing KIR2DL3, whereas cells expressing allomorphs, such as HLA-C*03:04 had a more limited capacity to inhibit NK cell activation. While the residues that contact KIR2DL2/3 were strictly conserved across all C1 allotypes assessed,

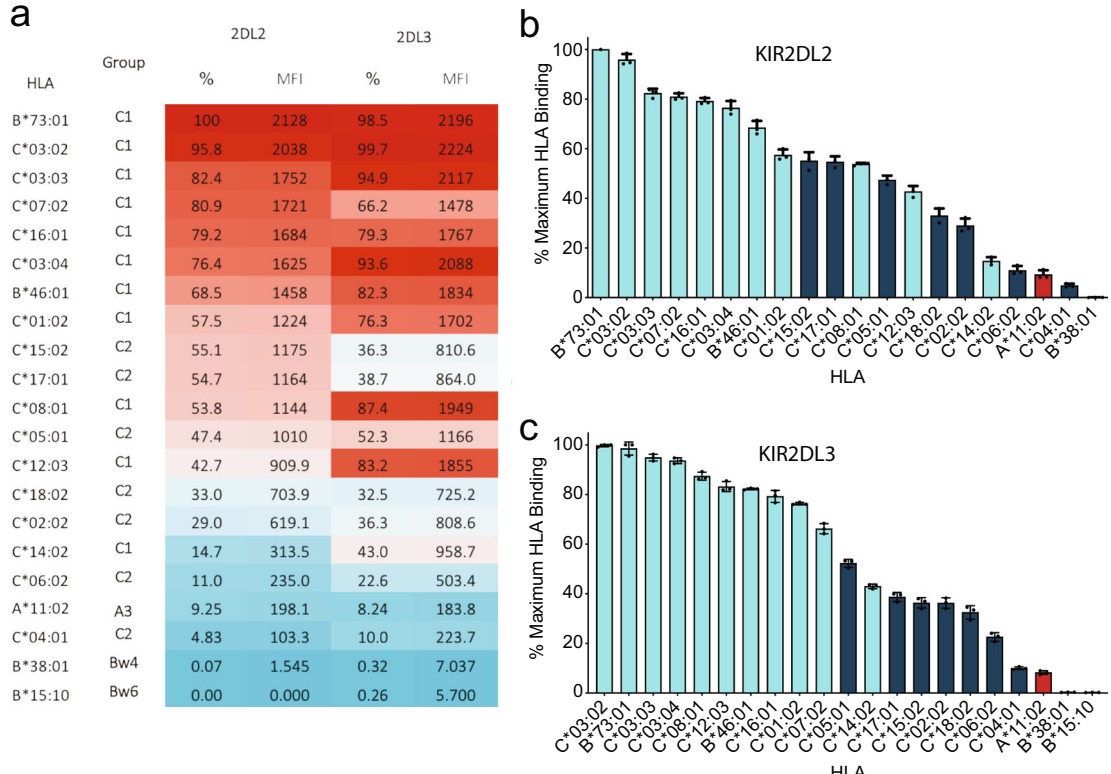

**Fig. 4 Single-antigen bead binding of KIR2DL2 and KIR2DL3.** Binding of KIR2DL2 and 2DL3 tetramers to a panel of 97 HLA class I molecules. **a** Shown as a heatmap by mean fluorescence intensity and percentage of maximum HLA-I ligand binding. **b** KIR2DL2 and **c** 2DL3 binding to individual HLA-I beads represented in bar format with errors displayed as standard deviation. HLA are categorised by KIR-binding epitope, HLA-C1 (cyan), HLA-C2 (dark blue), HLA A3/A11 (red). All data are representative of three independent experiments. The top 21 HLA ligands above background are shown. Error bars are shown as mean with SD.

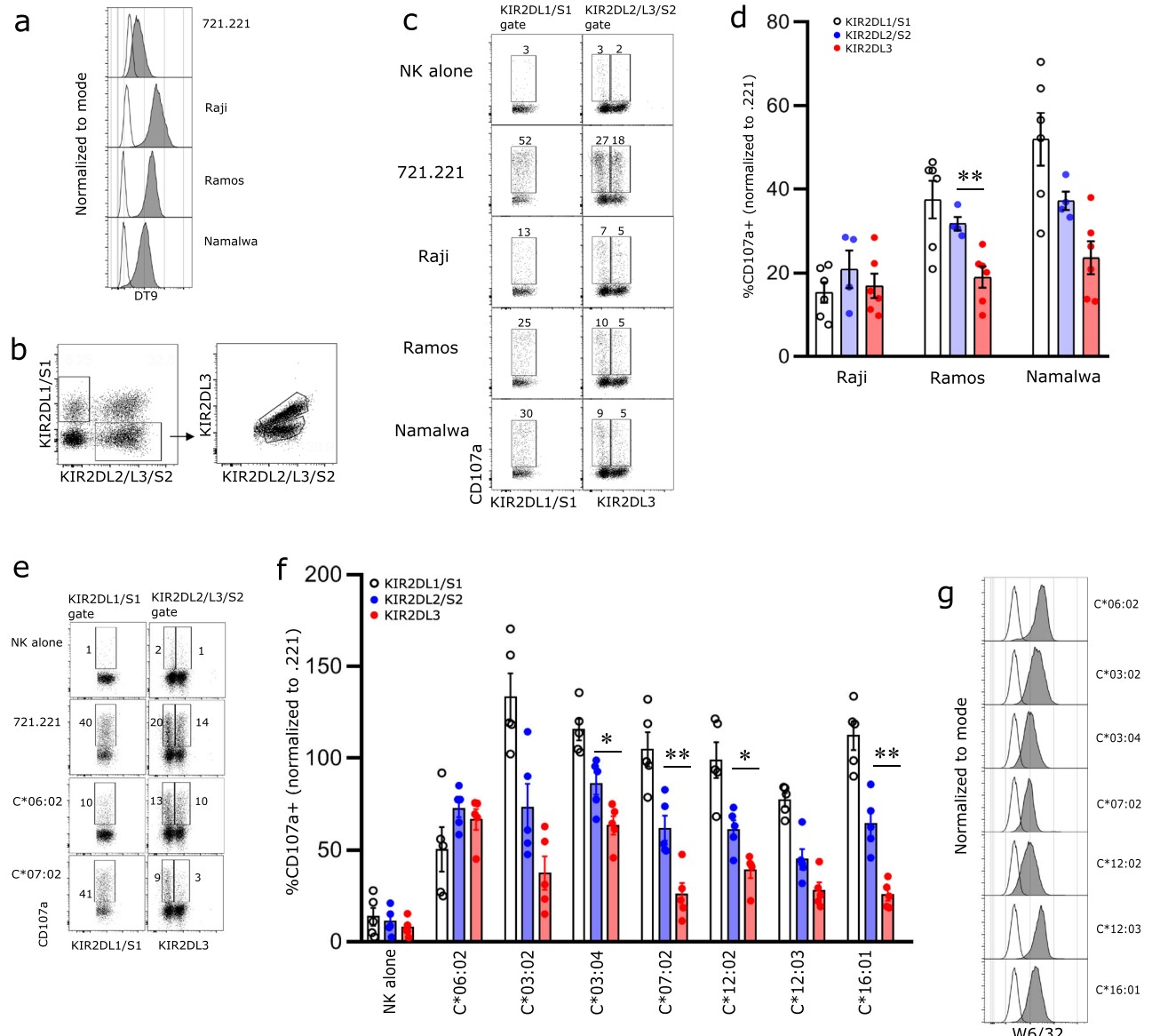

**Fig. 5 The functional response of KIR2DL2 and KIR2DL3 NK cells in the presence of HLA ligands.** Enriched NK cells from healthy donors were cultured with target cells and the degranulation response of NK cells expressing KIR2D assessed by flow cytometry. **a** The expression of HLA-C on both 721.221 and Raji, Ramos and Namalwa lymphoma cell lines following staining with DT9 antibody (shaded grey area) is shown, with secondary antibody alone shown as unfilled histogram. **b** Gating strategy for KIR2DL1/S1, KIR2DL2/S2 and KIR2DL3 NK cells. **c** Representative flow cytometry analysis of CD107a expression on NK cell subsets from one donor in the presence of 721.221, Raji, Ramos and Namalwa cells is shown (gated on KIR3DL1⁻ NK cells). **d** Pooled data showing CD107a+ cells from six donors with six KIR2DL1/S1, and KIR2DL3 populations and four KIR2DL2/S2 populations is shown. Each symbol represents an individual population of cells and bars depict mean ± SEM values. CD107a response is normalised to 721.221 response from the same experiment. **e** Representative CD107a expression on NK cell subsets from one donor in the presence of either 721.221 cells or 721.221 cells transfected HLA-C*06:02 or -C*07:02. **f** The degranulation response of KIR2D⁺ NK cells (GL183+ gated on a set GL183 MFI) to untransfected and HLA-C transfected 721.221 cells. Data are pooled from five donors. Each symbol represents data from an individual donor and bars depict mean ± SEM values. CD107a response is normalised to 721.221 response. **g** The expression of different HLA-C on target cells stained with W6/32 antibody (shaded grey area) is shown with 721.221 stain shown as black line. Statistical significance between KIR3DL2+ and KIR2DL3+ subsets (excluding KIR2DL1+) was tested using Mann–Whitney test for all conditions; no symbol = $p > 0.05$; *$p < 0.05$; **$p < 0.01$. Each experiment has been performed twice with data from both experiments pooled for the figure.

differences in their recognition did not strictly correlate with the level of HLA-C expression (Fig. 5g). This suggested that other allotypic features of these C1 molecules must also contribute to the extent of their recognition by NK cells. Similarly, the differences in inhibitory potency of different C1 allotypes did not strictly correlate with the binding data from the Luminex analyses. This in part may again reflect differences in the level of C1 allotype expression on the transfected 221 cells but also reflect complex patterns of co-

expression of receptors, such as KIR2DS2 which is typically present in conjunction with KIR2DL2 on B haplotypes which also have the potential to recognise C1 allotypes.

Consequently, to directly compare the response of KIR2DL2⁺ cells in the absence of KIR2DS2 with those that expressed KIR2DL3, we made use of the 1F12 mAb previously shown to recognise KIR2DL3 and KIR2DS2[33]. NK cells from donors who were homozygous for KIR2DL3 or KIR2DL2/S2 were initially

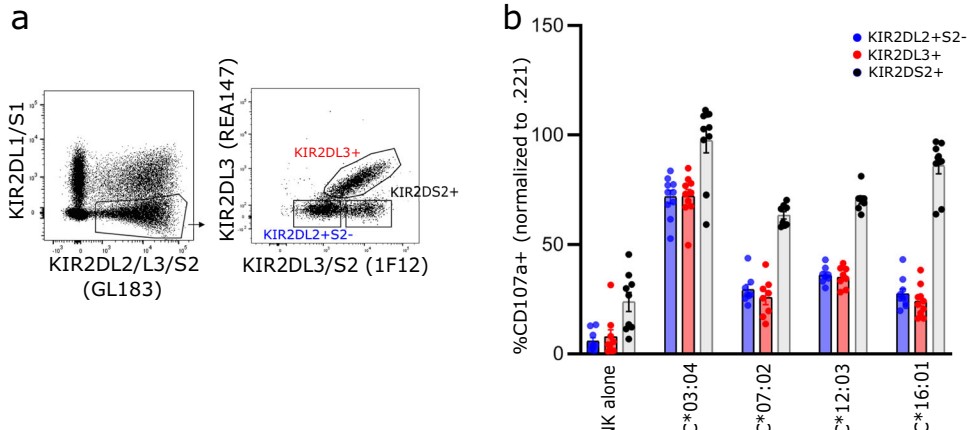

**Fig. 6 The functional response of KIR2DL2⁺S2⁻ and KIR2DL3 NK cells. a** Flow cytometry profile from one donor demonstrating gating strategy using 1F12 antibody. **b** Normalised responses of the gated populations following coculture either wild-type 221 cells or 221 cells transfected with HLA-C*03:04, -C*07:02, -C*12:03 or -C*16:01 is shown. Data are pooled from eight to ten donors from two experiments. Each symbol represents an individual donor and bars depict mean ± SEM values.

stained with GL183 (KIR2DL2/L3/S2), REA147(KIR2DL3) and 1F12 (KIR2DL3/S2) demonstrating that KIR2DL3⁺ and KIR2LD2⁺KIR2DS2⁻ populations could be discriminated (Supplementary Fig. 8). These subsets were also readily observed in individuals who possessed both KIR2DL2 and KIR2DL3 (Fig. 6a, Supplementary Fig. 8). The response of KIR2DL2⁺S2⁻L3⁻ and KIR2DL3⁺ subsets along with cells that expressed KIR2DS2 was then compared in heterozygous donors. While each of these populations had similar responses when stimulated with wild-type 221 cells, those that expressed KIR2DS2 exhibited elevated levels of activation relative to those that lacked KIR2DS2 upon coculture with .221-HLA-C transfectants (Fig. 6b, Supplementary Fig. 8). Indeed activation of the KIR2DS2⁺ population following stimulation with 221-HLA-C*03:04 and -C*16:01 was similar to that induced by untransfected 221 cells whereas cells expressing HLA-C*07:02 or -C*12:03 stimulated similar but slightly weaker responses. In contrast, the exclusion of KIR2DS2⁺ cells from the KIR2DL2 gate, revealed that KIR2DL2⁺/2DS2⁻ subset had a similar pattern of C1 allotype recognition to the KIR2DL3⁺ subset against each HLA-C allotype tested. In some donors there appeared to be preferential inhibition of KIR2DL3⁺ subset in response to HLA-C*07:02 and HLA-C*16:01, but this was not seen for all individuals perhaps reflecting allelic variation within the *KIR2DL2/3* genes. Furthermore, these similar patterns of C1-induced inhibition between KIR2DL2⁺/S2⁻ and KIR3DL3⁺ NK cells were also evident in donors who were homozygous for either KIR2DL2/S2 or KIR2DL3 (and lacking KIR2DS2), (Supplementary Fig. 8).

Again, the extent of inhibition varied with HLA-C allotype consistent with the potential of these receptors to subtly modulate the way they dock on different HLA-C/peptide complexes. Together, the data suggest that despite differences in docking mode, the functional response of NK cells expressing either KIR2DL2 or KIR2DL3 to cells bearing a range of different HLA-C allotypes was similar. This contrasted somewhat with the data from the analyses of KIR2DL2/3 binding to bead-bound HLA-C suggesting possible threshold effects above which subtle differences in the strength of interaction play at best a modest role in determining the extent of inhibition.

## Discussion
The interaction of KIR with their cognate HLA-I ligands plays a dual role in NK cell function, acting to educate NK cells and

facilitate discrimination between healthy and abnormal cells. In contrast to TCR recognition of HLA-I proteins that is typically highly specific for particular HLA-I allotypes[34], individual KIR recognise multiple HLA-I allotypes that share conserved motifs. While the structures of a number of KIR complexed to individual HLA-I ligands have been determined[28], it remained unclear whether any given KIR interact with multiple HLA-I ligands in precisely the same way, or whether different KIR that share HLA-I ligands, recognise them in the same way. To address this, the structures of KIR2DL2 complexed with HLA-C*07:02, and KIR2DL3 complexed to HLA-C*07:02 were determined. Notably, while KIR2DL2 and -2DL3 used essentially identical residues to contact both HLA-I ligands, there were nevertheless differences in the way each KIR2DL recognised HLA-C*03:04 and HLA-C*07:02. This finding was supported by our mutational analysis that showed KIR2DL2 and -2DL3 utilised a conserved constellation of residues to bind different regions on pHLA-C complexes. This unexpected capacity to use identical residues to make alternate pHLA contacts appears to be driven by differences in the juxtapositioning and specifically the twist of the D1 and D2 domains, and not by the D1-D2 interdomain angle as previously suggested[9,12]. This sort of plasticity is somewhat similar to that exhibited by the activating NK cell receptor NKG2D which recognises a range of MHC class-like ligands, binding to them with broadly similar docking modes, yet using similar residues in the receptor to make quite distinct molecular contacts with different ligands[35]. Critically, this plasticity has the potential to create a myriad of subtly distinct interaction modes of KIR2DL2 and KIR2DL3 with HLA-C peptide ligands. This likely impacts the capacity of each to accommodate variations in the HLA-bound peptide and ultimately drives differences in the recognition of the complex array of ligands expressed on the cell surface by each receptor.

SPR analyses of the KIR2DL interaction with HLA-C*07:02 suggested that the affinity for each HLA-C/peptide complex was broadly similar between KIR2DL2 and KIR2DL3. Further, while substitutions at positions 7 and 8 clearly impacted the affinity of these interactions, for the HLA-C*07:02-RL9 complex, they had similar impact on the binding of both KIR2DL2 and 2DL3. Whether or not this is a general phenomenon or a particular feature of the HLA-C*07:02-RL9 remains to be determined. Nevertheless, the data indicate that the affinity of KIR2DL2 for its ligands is not strictly of higher affinity than that of KIR2DL3.

In bead-binding assays and tetramer staining, we observed no general pattern of binding superiority by KIR2DL2. This contrasted with previous studies using KIR2DL-Ig fusion proteins which suggested that KIR2DL2 was a stronger and more peptide tolerant receptor than KIR2DL3[12,36–38]. There are significant intrinsic differences between the refolded KIR2DL2/3 compared with dimeric-KIR-Ig fusion proteins that include differences in glycosylation and the oligomerisation of the receptor which may impact docking geometries and/or modify the relative positioning of the D1 and D2 domains when bound to HLA-C. The use of tetrameric forms of KIR to probe the broader HLA specificity of these receptors similarly found no systematic preferential binding of KIR2DL2 to C1 allotypes over KIR2DL3. Rather, KIR2DL2 bound select allotypes, such as HLA-C*15:02 and -C*17:01 better than KIR2DL3 which in turn had stronger interactions with HLA-C*08:01 and -C*12:03 than KIR2DL2. Critically, given the potential of peptide variation to impact KIR binding, these broader specificity analyses of KIR2DL2 and -2DL3 used bead-bound HLA-C complexed with a cellular-derived peptide repertoire rather than a single peptide. Thus while KIR2DL2 may bind some HLA-C/peptide combinations better than KIR2DL3, our data suggest that are also HLA-C/peptide combinations that will be preferentially recognised by KIR2DL3. Moreover, given the potential of peptide to modulate recognition, whether or not a particular HLA-C allotype is preferentially bound by KIR2DL2 or KIR2DL3 is likely dependent on the cellular peptide repertoire and by implication the type of cell and potentially the extent to which its repertoire is modified by transformation and/or infection.

To better understand the extent to which these differences in HLA-C binding impacted target cell recognition by NK cells expressing KIR2DL2 or KIR2DL3, recognition of both tumour cells and transfected cell lines expressing different HLA-C allotypes was assessed. Our initial analyses again highlighted that HLA-C allotypes differed in their capacity to inhibit primary NK cells. Interestingly, the extent of inhibition observed did not correlate with cell surface levels of HLA-C suggesting that allotypic differences in HLA-C or in their associated peptide repertoires impacted recognition in much the same way as had been observed in binding analyses. However, the hierarchy of HLA-C allotype recognition observed in our functional analyses did not strictly correlate with that from the direct binding analyses. While the binding of the recombinant KIR2DL2/3 to HLA-coated beads exhibited the expected specificity for C1 allotypes, the extent to which modest differences between allotypes would be manifest in more physiological recognition settings is still unclear. For example, consistent with previous work, both binding and in vitro assays of cellular recognition indicate the HLA-C*07:02 is a strong ligand for both KIR2DL2 and KIR2DL3, yet exactly where it sits in the hierarchy of C1 allotypes is likely is impacted by numerous factors. For example, both the nature of the peptide repertoire and density of each HLA-C allotype present on the beads may significantly impact the extent of KIR tetramer binding. Indeed, the processes of purifying each HLA allotype and then generating stably coated beads might ultimately bias the bead-associated HLA-C to display a repertoire of peptides that has exceptional capacity to stabilise the HLA relative to that present on the cell surface. Similarly, the cellular analyses are also likely acutely dependent on both the peptide repertoire present on the surface of the target cell as well as the overall level of HLA-C expression. Indeed, overexpression of the HLA-C allotypes as is likely in the transfected 721.221 cells may well act to compress differences in recognition by KIR2DL2/3. Similarly, in the functional analyses NK cell intrinsic factors, such as differences in the KIR2DL2/3 allotypes expressed, the co-expression of activating KIR2DS, and education likely also contribute to functional outcomes, albeit that

similar responses were observed to 221 cells in all the donors who possessed C1 alleles not only in KIR2DL2+/S2− and KIR2DL3+ cells but by those who expressed KIR2DS2.

The data here also provided some comparison of the relative abilities of cells expressing either KIR2DL2 or KIR2DL3 to recognise a range of HLA-C allotypes. Our initial analyses comparing the extent of C1 inhibition between cells expressing KIR2DL3 with those that expressed either KIR2DL2 and/or KIR2DS2 suggested it was greater in the KIR2DL3+ population. However, upon exclusion of KIR2DS2+ cells, the responses of KIR2DL2+2DS2- cells to 721.221 cells expressing HLA-C*03:04, -C*07:02, -C*12:03 and -C*16:01 were each very similar to those that expressed KIR2DL3. In contrast, cells expressing KIR2DS2 while having similar responses to both KIR2DL2+/S2− and KIR2DL3+ cells to untranfected 721.221 cells, exhibited significant degranulation responses to cells transfected with C1 allotypes, albeit that they were not greater than that elicited by the parental 721.221 cells.

Taken together, our data highlight the potential for polymorphism in both receptor and ligand to regulate their mode of interaction. Indeed, the data imply that allotypic differences in the receptors, their HLA-I ligands and their associated peptide cargo may each impact the interaction and ultimately the quality of inhibitory signalling. Moreover, the impact of this genetically-driven variation in the quality of KIR2D/HLA-I interaction is essentially impossible to predict through sequence analysis alone. Thus, studies that further refine the impact of polymorphism in both receptors and ligands and that also account for the frequency of cells expressing both these and genetically linked receptors, such as KIR2DS2 will be needed to better understand how this intricate network of innate receptor/ligand combinations contributes to the control of infectious disease and cancer.

## Methods

**Cloning and expression of KIR2DL2 and KIR2DL3**. The extracellular D1-D2 domains (residues 1–204) of KIR2DL2*001 and KIR2DL3*001 were cloned into the expression vector pET-30(b) for expression in *E. coli* BL21 (DE3). The receptors were expressed as inclusion bodies and refolded and purified[24,39]. Briefly, 100 mg of the KIR2DL1, -L2 and -L3 were refolded by rapid dilution in a buffer containing 100 mM Tris-HCl pH 8.0, 400 mM L-arginine-HCl, 5 mM reduced glutathione, and 0.5 mM oxidised glutathione for 72 h. The refolded receptors were then applied onto a diethylaminoethyl (DEAE) cellulose column followed by size exclusion chromatography using Superdex 200 16/60 column (GE Healthcare). The KIR were further purified by anion exchange chromatography using Hi-Trap Q HP 5 ml column (GE Healthcare).

**Cloning and expression of HLA-C*03:04 and -C*07:02**. The extracellular $\alpha_1$, $\alpha_2$ and $\alpha_3$ domains of HLA-C*03:04 and -C*07:02 (residues 1–276) and human β2-macroglobulin (β2M) (residues 1–99) were cloned into the vector pET-30(b) for expression in *E. coli* BL21 (DE3). HLA-C*03:04 and C*07:02 and β2M were expressed separately into inclusion bodies and refolded in the presence of the GAVDPLLAL and RYRPGTVAL peptides, respectively[24,40]. Overall, 90 mg of the HLA heavy chain was refolded by rapid dilution in a buffer containing 100 mM Tris-HCl pH 8.0, 400 mM L-arginine-HCl, 5 mM reduced glutathione and 0.5 mM oxidised glutathione for 24 h, in the presence of 30 mg of β2 m and 10 mg of peptide. The refolded receptors were then applied onto a DEAE cellulose column followed by size exclusion chromatography using Superdex 200 16/60 column (GE Healthcare). The HLA were further purified by anion exchange chromatography using Hi-Trap Q HP 5 ml column (GE Healthcare). BirA-tagged HLA-C were generated by this refolding method and enzymatically biotinylated with biotin ligase for the generation of HLA-C tetramers[41].

**Surface plasmon resonance**. SPR experiments were performed on a BIAcore T3000 machine. Three independent experiments using different batches of protein were performed at 298 K in a buffer containing 10 mM HEPES pH 7.4, 150 mM NaCl and 0.05% Tween 20. The HLA-Class I specific monoclonal antibody W6/32 was coupled to four flow cells of a CM5 sensorchip (BIAcore) by amine coupling (~1000 resonance units (RU)). HLA-C*07:02 and alanine scanning mutants were captured by W6/32 to a density of ~150–200 RU. KIR2DL2 and 2DL3 were serially diluted in 10 mM HEPES pH 7.4, 150 mM NaCl, 0.05% Tween 20 (0–50 μM) and passed simultaneously over the test (HLA-C) and control (w6/32 alone) flow cells surfaces at a flow rate of 30 μl/min, with measurements taken in duplicate. The

obtained data were analysed using Prism (GraphPad) by steady-state equilibrium analysis following subtraction of control surface.

**Crystallisation, data collection, structure determination and refinement.** KIR2DL2 and KIR2DL3 were each concentrated to 10 mg/ml and added separately to HLA-C*07:02-RL9 at a 1:1 molar ratio. Crystals were obtained at 20 °C using the hanging drop vapour-diffusion method. Crystals of the KIR2DL2-HLA-C*07:02-RL9 complex were obtained from a solution consisting 0.2 M Sodium tartrate dibasic dihydrate and 20% w/v PEG 3350. Crystals of the KIR2DL3-HLA-C*07:02-RL9 complex were obtained from a solution containing 0.2 M Ammonium acetate, 0.1 M Tris-HCl pH.8.0 and 25% PEG 3350. Prior to flash-cooling in liquid nitrogen at 100 K, all crystals were equilibrated in a cryoprotectant solution containing their respective crystallisation solution and 35% PEG 3350. Data sets for the KIR2DL2 and KIR2DL3-HLA-C*07:02-RL9 complexes were collected to 3.1 Å and 2.5 Å respectively at the MX1 beamline (Australian Synchrotron, Victoria). The data were recorded on a Quantum-315 CCD detector and integrated and scaled using the HKL/HKL-2000 programme package[42] (data collection statistics are summarised in Supplementary Table 1). Structural determination proceeded by molecular replacement using PHASER, within the CCP4 programme suite[43] with the previously determined KIR2DL2-HLA-C*03:04 structure utilised as the search model (PDB accession code: 1EFX[24]). Both the KIR2DL2 and 2DL3 crystals comprised two copies of their respective complexes in the asymmetric unit. Refinement of the complexes progressed with iterative rounds of manual building in COOT[44] and refinement in PHENIX[45] with twofold non-crystallographic symmetry restraints applied throughout. The final model of the KIR2DL2-HLA-C*07:02-RL9 complex comprises residues 4–190. The final model of the KIR2DL3-HLA-C*07:02-RL9 complex comprises residues 6–195. The structures were validated with MOLPROBITY[46]. The KIR2DL2 and 2DL3-HLA-C*07:02-RL9 complexes were deposited in the Protein Databank under accession codes 6PA1 and 6PAG, respectively (https://doi.org/10.2210/pdb6PA1/pdb and https://doi.org/10.2210/pdb6PAG/pdb).

**KIR2DL2 and 2DL3 mutagenesis and transfection.** Full-length constructs of KIR2DL2 and KIR2DL3 were cloned into the pEF6 vector with a FLAG tag at the N-terminus of the coding sequence. Site-directed mutagenesis (QuikChange II, Stratagene) was used to generate mutants of KIR2DL2 and KIR2DL3, in order to study the contribution of particular amino acids located on their ectodomains to HLA-I recognition (Supplementary Table 3). Transfection of all constructs into HEK293T cells was accomplished using FuGENE 6 transfection reagent (Roche) based on the manufacturer's instructions. HEK293T cell transfectants expressing wild-type or mutant KIR2DL2 and KIR2DL3 were stained with HLA-C tetramers and/or anti-FLAG (clone M2; Sigma-Aldrich) mAb for 30 min at 277 K and then analysed by flow cytometry.

**Single-antigen bead assay.** HLA-I recognition by KIR2DL2 and 2DL3 and mutants thereof was assessed through the use of a multiplexed bead assay. Briefly, 5 μg of PE-conjugated KIR tetramers were incubated for 30 min at room temperature with beads coated with a panel of 100 different HLA-A, B and C molecules (LABScreen HLA Class I Single Antigen; One Lambda). Samples were processed according to the manufacturer's instructions using LABScreen Wash Buffer (One Lambda) and binding measured on a Luminex platform (LABScan 100; One Lambda). Interactions with individual HLA allotypes were distinguished via unique bead labelling and detection of tetramer fluorescence intensity. Normalised fluorescence values were obtained using the HLA Fusion software suite (One Lambda) using Eq. 1:

$$(S\#N - SNC\ bead) - (BG\#N - BGNC\ bead), \tag{1}$$

where S#N is the sample-specific fluorescence value (trimmed mean) for bead #N, SNC bead is the sample-specific fluorescence value for the negative control (nude) bead, BG#N is the background negative control fluorescence value for bead #N, and BGNC bead is the background negative control fluorescence value for negative control bead. Negative control samples were obtained using an isotype control (PE-conjugated anti-human IgG –One Lambda) and were subtracted from the raw values obtained for each experiment. Recognition was assessed as being higher than that of the tetramer to beads without conjugated HLA-I. MFI values were normalised to the highest value for each experiment and then averaged over three experiments.

**Degranulation assays.** Cryopreserved PBMCs from healthy blood bank donors were thawed and washed several times before NK cells were enriched using EasySep human NK enrichment kit (Stemcell Technologies) as per the manufacturer's instructions. NK cells were cultured overnight in 100 U/ml recombinant human IL-2 (IS premium grade, Miltenyi Biotec). NK cells were incubated with target cells at a 1:1 ratio for 5 h. In all experiments, CD94-NKG2A was blocked with supernatant from the Y9 hybridoma (kind gift from Gabriella Pietra, University of Genova, Italy) added 30 min before the assay, and maintained in the culture throughout the assay. Antibody to CD107a PE, PECy5 or PECy7 (clone H4A3; BD Biosciences; at 1:100 final dilution) was added at the beginning of the culture followed by Golgi Stop (Monensin, BD Biosciences). The following cell

lines were used as targets for functional assays: HLA-deficient B lymphoblastoid cell line 721.221 (.221); .221 cells stably transfected with HLA-C1 and C2; Burkitt's lymphoma cell lines used were Raji (HLA-A*03:01, B*15:10, C*03:04/C*04:01), Ramos (HLA-A*03:01, B*44:160Q/51:01, C*16:01) and Namalwa (HLA-A*03:01/68:02, B*07:02, B*49:01, C*07:01/07:02) sequenced by Victorian Transplantation and Immunogenetics Service, Melbourne, Australia. The experiments were performed in accordance with approvals from the Human Research Ethics Committee, University of Melbourne.

**Flow cytometry.** Single-cell suspension was incubated in staining antibody cocktail containing a combination of following antibodies: CD56 BV421 (clone NCAM16.2; BD Biosciences; 1:50 dilution), CD56 APC or PE (clone B159; BD Biosciences; 1:50 dilution), KIR3DL1 APC (clone DX9; BD Biosciences; 1:50 dilution), CD3 APCCy7 (clone SK7; BD Biosciences; 1:100 dilution), KIR2DL2/L3/S2 PECy5.5 (clone GL183; Beckman Coulter; 1:50-1:200 dilution), KIR2DL2/L3/S2 BUV737 (clone CH-L; BD Biosciences; 1:50 dilution), KIR2DL1/S1 PECy7 or APC (clone EB6; Beckman Coulter; 1:50 dilution), KIR2DL3 FITC (clone REA147; Miltenyi Biotech; 1:50 dilution), fixable viability dye eFluor780 (Life Research; 1:500 dilution). Cells were surface stained in FACS buffer (PBS containing 2% FCS and 5 mM EDTA) on ice, in dark for 30 min after which they were washed twice and fixed using BD Cytofix/Cytoperm kit (BD Pharmingen). Staining with the KIR2DL3/S2 specific antibody 1F12 (EFS, Nantes, France) employed a multi-step protocol with a secondary antibody and was performed as follows: cells were initially incubated with a goat anti-mouse IgG antibody (polyclonal; Invitrogen) for 30 min on ice to block secondary Ab-binding sites on cell-bound CD107a-specific mAb before incubation with purified mouse anti-human 1F12 antibody (1:50 dilution), followed by secondary goat anti-mouse IgG BV421 antibody (clone poly4053; Biolegend; 1:50 dilution). After washing, cells were incubated in FACS buffer with normal mouse serum (Thermo Fisher Scientific) before adding the remaining antibody cocktail and fixing the cells.

All flow cytometry experiments were performed on Flow Cytometers BD LSRII or BD LSR Fortessa. Analysis was performed using FlowJo software (Tree Star). Gating strategy included the elimination of dead cells positive for fixable viability stain, CD3 antibody stain and doublets, including the exclusion of cells based on size and granularity using FSC/SSC discrimination. NK cells were identified as viable, CD3-lymphocytes, positive for CD56. Target cells were analysed for the expression of HLA I (clone w6/32) and HLA-C (clone DT9; Sigma; dilution 1:50). Secondary antibody chicken anti-mouse IgG AF488 (Life Technologies; dilution 1:100) was used to identify a positive stain. Gating strategies for NK cell subset analyses are provided (Supplementary Fig. 9).

**Reporting summary.** Further information on experimental design is available in the Nature Research Reporting Summary linked to this paper.

## Data availability
All data are available upon request from the authors. Crystal structures of the KIR2DL2 and KIR2DL3-HLA-C*07:02-RL9 complexes were deposited in the Protein Databank under accession codes 6PA1 and 6PAG, respectively. Source data are provided with this paper.

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

## Author contributions

S.M., J.P.V. and P.P collected and analysed the structural and bead binding data. S.S, G.M.C, J.W., L.C.S, S.C.L., C.R. and P.M.S collected and analysed the cellular and functional data. B.J.M and C.F. collected and analysed the SPR data. J.R., A.G.B and J.P.V conceived and designed the study. All authors contributed to the writing of the manuscript.

## Competing interests

The authors declare no competing interests.
