## [Peer Review File · Nature Communications]

Reviewers' comments:

Reviewer #1 (Remarks to the Author):

Moradi et al

The KIR:HLA system is a complex receptor:ligand system that controls NK cells. Genetic diversity is generated at the level of the locus as well as that at the allelic level. This has been the subject of a number of disease association studies and the authors have attempted to define a structural basis for these differences. They have studied KIR2DL2 and KIR2DL3 which have similar (but not identical ligands) and C*0702 and C*0304 ligands. Whilst they have focussed structural efforts on HLA-C*0702 with the peptide RYRPGTVAL, demonstrating subtle difference in engagement of KIR2DL2 and KIR2DL3 with this peptide:MHC complex. In SPR studies the affinities appear comparable for this complex, although for the comparison of KIR2DL2 and KIR2DL3 with HLA-C*0304 and GAVPDLLAL, KIR2DL3 had a higher affinity for the peptide:MHC complex. They have performed additional binding and functional experiments to test the differences between KIR2DL2 and KIR2DL3. The work is important because understanding diversity in the KIR:MHC interaction can provide a molecular rationale for the findings from immunogenetic disease association studies. The main novelty of the work lies in the structural analysis which is detailed and interesting, but some previous work (eg Moesta et al J Immunol 2008 and Sim et al Frontiers in Immunology 2017) have shown that KIR2DL2 and KIR2DL3 have subtly different peptide:MHC specificities

Comments

The structural studies set out the basis for the functional work. The study of multiple different HLA types is interesting. However there is additional complexity in that KIR2DL2 in general binds more HLA-C allotypes and has a broader peptide tolerance than KIR2DL3. Whilst the authors note that there is diversity in the binding of KIR2DL2 and KIR2DL3 to different HLA-C molecules it is not clear why this impacts disease.

The correlation between structure and function is not clear. For instance, in Figure 6 the KIR2DL2/S2 positive NK cells are less inhibited than the KIR2DL3-positive NK cells against both HLA-C*0702 and HLA-C*0304. However in the SPR assays the two KIR bind HLA-C*0702 with similar affinities, but HLA-C*0304 with different affinities?

It is possible that this related to the method of identifying the different sub-groups. For instance they have used the antibody RAE147 (I think they mean REA147) to identify KIR2DL3+ NK cells. This is not robust as this Ab also picks up KIR2DL2 (Ref Blunt et al HLA 2019). Within their function experiments they are also picking up KIR2DS2 with the GL183 Ab in addition to KIR2DL2 and KIR2DL3. Thus differences in function which the authors have attributed to differences in KIR2DL2 and KIR2DL3, may in fact be related to KIR2DS2 which may lead to activation. Thus, overall the experiments in Figure 6 do not accurately represent single KIR-positive NK cells and hence the results of these experiments are not robust in terms of interpretation in the context of the structural data. These experiments need repeating with more robust identification of KIR2DL2 and KIR2DL3 eg a combination of genetics and staining or cloning.

Previous work has shown that the KIR are peptide selective. How does their choice of peptide influence the work?

Figure 4: is confusing as the asterisks do not apply to pairs of columns, but to comparisons between panels A and B.

Figure 6: The data are normalised to reactivity of 721.21. Are there any differences in the levels of activity against this class I negative cell line? Does education of NK cells affect

reactivity eg donor HLA-C genotype? IT has been previously suggested that HLA-C*07 is a “strong educator” ie high levels of baseline activity against a class I negative cell line.

Reviewer #2 (Remarks to the Author):

Moradi et al. determined the crystal structures of HLA-Cw*07:02 complexed either with natural killer cell receptor KIR2DL2 or KIR2DL3, both of which have been believed to be specific for HLA-C1 group. They revealed similar but slightly different binding modes of these KIRs toward the same HLA-Cw*07:02. They further performed the mutagenesis study on KIRs expressed on 293 cells to identify some amino acids responsible for HLA binding and specificity. On the other hand, they also analyzed the binding specificity of KIR2DL2 and KIR2DL3 tetramers toward the beads attached with 18 HLA allele proteins and cells. KIR2DL3 showed more specific binding to HLA-C1 group than KIR2DL2. Furthermore, they evaluated the activation of subsets of NK cells divided by expression levels of specific KIRs, again showing that KIR2DL3-expressing NK cells was more inhibited by HLA-C1 group than KIR2DL2-expressing ones. These results provide insights on the functional and structural differences of HLA-C specific KIR2Ds. This paper is interesting and has some potential to advance our knowledge for a field of NK cell immunology, but they should discuss more about related reports such as Moesta, A.K. et al. *J Immunol* 180, 3969-79 (2008). I have some comments.

Comments

1. The authors should discuss the relationship of crystal structures of KIR2DL-HLA-C complexes with functional data of KIR mutants as well as HLA alleles, especially shown in Figures 5 and 6. Is it possible to explain these functional differences of HLA allele specificities from current structures?
2. Is there any clear correlation of functional data between Figures 5 and 6F? HLA-C1203 bound to KIR2DL3 only, while HLA-C1601 and C0304 did to both KIRs. However, the NK cell activation might not seem to be related to binding results.
3. In Figure 4, raw FCM data, at least representative data, should be presented. Did they check the expression levels of KIR mutants? How did they calculate % positive cells (more than 100% of positive cells)?

Minor comments

1. The electron density map of some area such as binding interfaces should be shown.
2. In Figure 3, the authors tried to show the equilibrium binding experiments. However, KIR2DL2-Cw0304 binding show decreasing responses at saturated states, and KIR2DL3-Cw3 and -Cw7 both increasing responses. These seem to have either inappropriate control or nonspecific binding. How did they determine the responses for fitting? In principle, they should perform the binding with short time of saturation states to reduce the effect of nonspecific binding.
3. In Supplementary Figure 1, Asp187 of both KIRs seem to be located distantly from the F-F' loops.
4. In Figure 2D, the differences of interactions of binding amino acids are hardly understood, maybe because coloring and color depth are similar.

Reviewer #3 (Remarks to the Author):

Understanding the molecular basis for genetic associations of KIR and their HLA class I ligands with disease is a key question in the field. Multiple associations have demonstrated a difference in disease risk between individuals who carry KIR2DL2 and KIR2DL3, two highly related receptors that bind the same ligand, HLA-C allotypes of group 1 (C1).

The manuscript by Moradi et al. describes novel crystal structures of KIR2DL2 and KIR2DL3 bound to the same HLA-C*07:02 ligand, allowing a direct comparison of ligand recognition. These structures and the accompanying tetramer binding studies that map the specific binding sites are novel contributions. However, the manuscript provides little on the question of why disease risk differs between individuals who carry KIR2DL2 and KIR2DL3.

The major conclusion that KIR2DL2 is a weaker receptor than KIR2DL3 and that this may be due to the differing KIR docking geometries is not consistent with the data presented. Notably, in the bead binding assay KIR2DL2 is a stronger receptor than KIR2DL3 for C*07:02, while by surface plasmon resonance they are the same. Thus, KIR2DL2 is not a weaker receptor than KIR2DL3 for C*07:02. The different docking geometries must be incidental to differences in binding strength.

The authors allude to the role of peptide sequence in modifying KIR recognition of HLA-C but do not cite key references on this topic. Binding of KIR2DL2 and KIR2DL3 to C1 allotypes is quite sensitive to the peptide sequence. A limitation of their structural analysis is that it is based on C*07:02 in the context of only one peptide sequence.

NK cell degranulation assays showed stronger inhibition by KIR2DL3+ compared to KIR2DL2+ NK cells, supporting their conclusion that KIR2DL2 is a weaker receptor than KIR2DL3. However, the interpretation of these experiments is significantly impacted by the inability to eliminate the contribution of the activating receptor KIR2DS2, since the antibody used does not discriminate between KIR2DL2 and KIR2DS2. NK cells that coexpress these two receptors may have diminished inhibition due to the contribution of KIR2DS2. Although the authors note this shortcoming, they do not consider it when drawing their conclusion.

Multiple publications have noted that KIR2DL3 binding to C1 allotypes is weaker than KIR2DL2 on cells or beads when using soluble KIR-Fc fusion proteins. The reason(s) for this weaker binding is still not understood. The authors used refolded KIR tetramers for their bead binding experiments and observed that KIR2DL2 binds more weakly to C1 allotypes than KIR2DL3. However, the authors did not resolve this discrepancy.

Reviewer #1 (Remarks to the Author):

Moradi et al

The KIR:HLA system is a complex receptor:ligand system that controls NK cells. Genetic diversity is generated at the level of the locus as well as that at the allelic level. This has been the subject of a number of disease association studies and the authors have attempted to define a structural basis for these differences. They have studied KIR2DL2 and KIR2DL3 which have similar (but not identical ligands) and C*0702 and C*0304 ligands. Whilst they have focussed structural efforts on HLA-C*0702 with the peptide RYRPGTV_{AL}, demonstrating subtle difference in engagement of KIR2DL2 and KIR2DL3 with this peptide:MHC complex. In SPR studies the affinities appear comparable for this complex, although for the comparison of KIR2DL2 and KIR2DL3 with HLA-C*0304 and GAVPDLL_{AL}, KIR2DL3 had a higher affinity for the peptide:MHC complex. They have performed additional binding and functional experiments to test the differences between KIR2DL2 and KIR2DL3. The work is important because understanding diversity in the KIR:MHC interaction can provide a molecular rationale for the findings from immunogenetic disease association studies. The main novelty of the work lies in the structural analysis which is detailed and interesting, but some previous work (eg Moesta et al J Immunol 2008 and Sim et al Frontiers in Immunology 2017) have shown that KIR2DL2 and KIR2DL3 have subtly different peptide:MHC specificities

Comments

The structural studies set out the basis for the functional work. The study of multiple different HLA types is interesting. However there is additional complexity in that KIR2DL2 in general binds more HLA-C allotypes and has a broader peptide tolerance than KIR2DL3. Whilst the authors note that there is diversity in the binding of KIR2DL2 and KIR2DL3 to different HLA-C molecules it is not clear why this impacts disease.

The correlation between structure and function is not clear. For instance, in Figure 6 the KIR2DL2/S2 positive NK cells are less inhibited than the KIR2DL3-positive NK cells against both HLA-C*0702 and HLA-C*0304. However in the SPR assays the two KIR bind HLA-C*0702 with similar affinities, but HLA-C*0304 with different affinities?

It is possible that this related to the method of identifying the different sub-groups. For instance they have used the antibody RAE147 (I think they mean REA147) to identify KIR2DL3+ NK cells. This is not robust as this Ab also picks up KIR2DL2 (Ref Blunt et al HLA 2019). Within their function experiments they are also picking up KIR2DS2 with the GL183 Ab in addition to KIR2DL2 and KIR2DL3. Thus differences in function which the authors have attributed to differences in KIR2DL2 and KIR2DL3, may in fact be related to KIR2DS2 which may lead to activation. Thus, overall the experiments in Figure 6 do not accurately represent single KIR-positive NK cells and hence the results of these experiments are not robust in terms of interpretation in the context of the structural data. These experiments need repeating with more robust identification of KIR2DL2 and KIR2DL3 eg a combination of genetics and staining or cloning.

Previous work has shown that the KIR are peptide selective. How does their choice of peptide influence the work?

Figure 4: is confusing as the asterisks do not apply to pairs of columns, but to comparisons between panels A and B.

Figure 6: The data are normalised to reactivity of 721.21. Are there any differences in the levels of activity against this class I negative cell line? Does education of NK cells affect reactivity eg donor HLA-C genotype? IT has been previously suggested that HLA-C*07 is a “strong educator” ie high levels of baseline activity against a class I negative cell line.

Response to Reviewer #1

We thank reviewer #1 for their careful critique and for stating that ‘**the work is important**’, and for recognising that the ‘**main novelty of the work lies in the structural analysis which is detailed and interesting**’. The reviewer made a number of comments, which we address in turn below.

“Moesta et al J Immunol 2008 and Sim et al Frontiers Immunol 2017 have shown that KIR2DL2 and KIR2DL3 have subtly different peptide:MHC specificities”

We agree with the reviewer that previous work has observed differences in peptide specificities between KIR2DL2 and KIR2DL3. In these instances, KIR2DL2 is the “stronger” receptor. Our structural analyses

show that KIR2DL2 and KIR2DL3 dock on their ligands using molecular mechanisms that are subtly distinct and that vary with individual HLA-peptide combinations. In keeping with this, our work here shows that KIR2DL2 is not necessarily a stronger receptor. We have added the following comment in the discussion (pages 11-12) to indicate that this structural variation can create perceived recognition preferences.

“Critically, this plasticity has the potential to create a myriad of subtly distinct interaction modes of KIR2DL2 and KIR2DL3 with HLA-C peptide ligands. This likely impacts the capacity of each to accommodate variations in the HLA-bound peptide and ultimately drives differences in the recognition of the complex array of ligands expressed on the cell surface by each receptor.”

“However, there is additional complexity in that KIR2DL2 generally binds more HLA-C allotypes and has a broader peptide repertoire tolerance than KIR2DL[3]”

Previous reports have suggested that KIR2DL2 is the more broadly tolerant receptor. We have noted these studies and observed that the most robust data supporting this contention is derived from studies using KIR-Ig fusion proteins. Consistent with the structural observations of plasticity in the KIR/C1 interaction, we find that while there are instances of KIR2DL2 interacting somewhat better with some HLA-C allotypes, others interacted slightly better with KIR2DL3. Similarly, our SPR analyses showed that the affinity of KIR2DL2 and KIR2DL3 for highly defined HLA-C/peptide complexes was very similar. We have addressed this in the discussion as follows (page 12):

*“In bead-binding assays and tetramer staining we observed no general pattern of binding superiority by KIR2DL2. This contrasted with previous studies using KIR2DL-Ig fusion proteins which suggested that KIR2DL2 was a stronger and more peptide tolerant receptor than KIR2DL3^{12,36-38}. There are significant intrinsic differences between the refolded KIR2DL2/3 compared with dimeric-KIR-Ig fusion proteins that include differences in glycosylation and the oligomerisation of the receptor which may impact docking geometries and/or modify the relative positioning of the D1 and D2 domains when bound to HLA-C. The use of tetrameric forms of KIR to probe the broader HLA specificity of these receptors similarly found no systematic preferential binding of KIR2DL2 to C1 allotypes over KIR2DL3. Rather, KIR2DL2 bound select allotypes such as HLA-C*15:02 and -C*17:01 better than KIR2DL3 which in turn had stronger interactions with HLA-C*08:01 and -C*12:03 than KIR2DL2. Critically, given the potential of peptide variation to impact KIR binding, these broader specificity analyses of KIR2DL2 and -2DL3 used bead*

bound HLA-C complexed with a peptide repertoire rather than a single peptide. Thus while KIR2DL2 may bind some HLA-C/peptide combinations better than KIR2DL3, our data suggests that are also HLA-C/peptide combinations that will be preferentially recognised by KIR2DL3. Moreover, given the potential of peptide to modulate recognition, whether or not a particular HLA-C allotype is preferentially bound by KIR2DL2 or KIR2DL3 is likely dependent on the cellular peptide repertoire and by implication the type of cell and potentially the extent to which its repertoire is modified by transformation and/or infection.”

“The authors note that there is diversity in the binding of KIR2DL2 and KIR2DL3 to ...HLA-C... not clear why this impacts disease”

The differences between KIR2DL2 and KIR2DL3 on disease progression have been described at the genetic association level (Khakoo et al Science 2004 and Romero et al Mol Immunol 2008). These studies have suggested the “weaker” inhibitory potential of KIR2DL3 provides a more robust immune response to HCV challenge. Our data suggest that KIR2DL3 is not necessarily a “weaker” receptor than KIR2DL2. Further, we show that KIR2DL3 and KIR2DL2 have different hierarchies of binding affinity for HLA-C alleles, and again KIR2DL3 is not necessarily the weaker binder. Thus, our results impact on the interpretation of the genetic association data. We have altered the concluding remarks of the discussion to now read (page 14):

“Thus, studies that further refine the impact of polymorphism in both receptors and ligands and that also account for the frequency of cells expressing both these and genetically linked receptors such as KIR2DS2 will be needed to better understand how this intricate network of innate receptor/ligand combinations contributes to the control of infectious disease and cancer.”

“...the KIR2DL2/S2 positive cells are less inhibited than the KIR2DL3 positive cells against both HLA-C*07:02 and C*03:04. However, in the SPR assays the two bind C*07:02 with similar affinities but C*03:04 with different affinities? Structure and SPR are single peptide studies”

Reviewer #1 correctly points out that single peptide studies such as SPR do not always reflect studies performed with cellular repertoires of peptides such as functional assays. Deconvoluting the contribution of HLA and KIR allelic differences from that of peptide remains a considerable obstacle in KIR research. We consider a strength of our study to be that we studied this system at both single peptide and peptide

repertoire levels. At both levels we observed differences in KIR2DL2 and KIR2DL3 binding. Notably, KIR2DL2 and KIR2DL3 structures were determined on the same HLA-peptide and yet revealed structural differences. Further, in luminex-based binding assays, which presents cellular peptide repertoires, clear differences between KIR2DL2 and 2DL3 binding were observed. We have now expanded our SPR analyses of the interaction between KIR2DL2/3 and HLA-C*07:02 to include 12 peptides in addition to the original RL9 peptide. This data is now included as Supplementary Figure 6.

“It is possible that this related to the method of identifying the different sub-groups. For instance they have used the antibody RAE147 (I think they mean REA147) to identify KIR2DL3+ NK cells. This is not robust as this Ab also picks up KIR2DL2 (Ref Blunt et al HLA 2019). Within their function experiments they are also picking up KIR2DS2 with the GL183 Ab in addition to KIR2DL2 and KIR2DL3. Thus differences in function which the authors have attributed to differences in KIR2DL2 and KIR2DL3, may in fact be related to KIR2DS2 which may lead to activation. Thus, overall the experiments in Figure 6 do not accurately represent single KIR-positive NK cells and hence the results of these experiments are not robust in terms of interpretation in the context of the structural data. These experiments need repeating with more robust identification of KIR2DL2 and KIR2DL3 eg a combination of genetics and staining or cloning.”

We thank the reviewer #1 in pointing out the error in name of the antibody clone. Indeed, we have used the antibody REA147, to identify KIR2DL3 +ve cells. We have carefully assessed the staining profile of the antibodies REA147, GL183 and CHL by staining both 293T cells transfected with plasmids encoding containing KIR2DL2 or KIR2DL3 as well as primary NK cells. In our hands REA147 can be used in conjunction with GL183 clone to reliably distinguish KIR2DL3+ cells from other GL183+ ve cells as seen in Supplementary Figure 7 where it has slight cross reactivity with KIR2DL2 but only on cells that express very high levels of KIR2DL2. Neither CHL nor GL183 could discriminate between these two receptors.

However careful titration of these mAbs only allowed for the discrimination of KIR2DL3 from KIR2DS2/L2. To more fully address the reviewer’s concerns regarding KIR2DS2, we have now included an additional mAb 1F12 in the staining cocktail which recognises KIR2DL3 and KIR2DS2 (David et al., Immunology 128:172, 2009). This data is now included as Figure 6 and Supplementary Figure 8. The combination of 1F12 and REA147 can be used to distinguish the two populations of

interest, namely KIR2DL2+KIR2DS2- from KIR2DL3+ cells. In heterozygous donors, cells that expressed KIR2DL2/L3/S2 identified by GL183 (gated) were subsequently assessed for reactivity with 1F12 and REA147. Those negative for both 1F12 or REA147 were identified as KIR2DL2+KIR2DS2-KIR2DL3- cells and typically comprised around 17% of GL183 positive cells in heterozygous donors. Cells that stained with 1F12 but were negative for REA147 expressed KIR2DS2 with or without KIR2DL2 (28%). Cells positive for REA147 were identified as KIR2DL3+ (46%). This staining pattern was validated by staining samples from individuals who were homozygous for either KIR2DL2 or KIR2DL3 (Supplementary Figure 8).

Using this refined gating strategy, we have now extended the functional analyses as requested, excluding cells that express KIR2DS2. The data showed that there are minimal differences between KIR2DL2 and KIR2DL3 NK cell subsets in their ability to recognise HLA-C (Figure 6B and Supplementary Fig 8). The figures/abstract and conclusions have been modified accordingly.

“Figure [5]: The data are normalised to reactivity of 721.21. Are there any differences in the levels of activity against this class I negative cell line? Does education of NK cells affect reactivity eg donor HLA-C genotype? IT has been previously suggested that HLA-C*07 is a “strong educator” ie high levels of baseline activity against a class I negative cell line.”

Primary cells used in this donor were all obtained from *C1*+ donors which is now indicated in the discussion as below. Indeed, with the exception of 1 donor, all possessed either HLA-C*07:01 or -C*07:02.

(Page 13, line 22)

“..., albeit that all the donors used in the analyses here possessed C1 alleles and similar responses were observed to 221 cells by not only KIR2DL2⁺/S2⁻ and KIR2DL3⁺ cells as well as those the expressed KIR2DS2.”

We have also now included the absolute responses of KIR2DL2+S2-, KIR2DL3+ and KIR2DS2+ cells to .221 cells in Supplementary figure 8 and referred to the data as above as well as in the results section as below (page 10, line 27):

While each of these populations had similar responses when stimulated with wild type 221 cells, those that expressed KIR2DS2 exhibited elevated levels of activation relative to those that lacked KIR2DS2 upon coculture with .221-HLA-C transfectants (Figure 6B and Supplementary Figure 8).

“How does their choice of peptide influence the work?”

The choice of peptide is paramount on two different levels. Firstly, the choice of peptide influences our ability to refold HLA-C at sufficient yields to enable downstream SPR and structural studies. Secondly, the choice of peptide is key with respect to the affinity of the KIR-HLA-peptide interaction, where high affinity interactions lends itself towards obtaining well diffracting crystals, thereby enabling structural determination. We broadened the SPR analysis with 14 substitutions across the P7 and P8 positions within HLA-C*07:02-RL9 complex. This is now reflected in the results section, Supplementary Figure 6 and discussion sections as indicated below (page 8, line 5).

*“As the above observations were based on KIR2DL recognition of HLA-C1 allotypes bound to a single peptide, we next examined the extent to which differences between KIR2DL2 and KIR2DL3 could be modulated by peptide repertoire. We measured KIR2DL2 and KIR2DL3 binding via SPR to a panel of twelve HLA-C*07:02-RL9 peptide substitutions which spanned a variety of amino acids in the key KIR2DL contact regions P7 and P8 (Supplementary Figure 6). With the exception of P8F and P8V, each substitution resulted in reduced affinity binding to both KIR2DL2 and KIR2DL3 (Supplementary Table 3). For example, inclusion of the acidic residue Glu at P7 or P8 was detrimental to binding of both KIR2DLs in line with previous findings³². However, both KIR2DL2 and KIR2DL3 shared similar binding preferences across the peptides in this panel (Supplementary Table 3). Thus, at least in the context of HLA-C*07:02-RL9 peptide, there were no marked differences between KIR2DL2 and 2DL3 with respect to their sensitivity to P7 and P8 substitutions.”*

And (page 12, line 6):

*“SPR analyses of the KIR2DL interaction with HLA-C*07:02 suggested that the affinity for each HLA-C/peptide complex was broadly similar between KIR2DL2 and KIR2DL3. Further, while substitutions at positions 7 and 8 clearly impacted the affinity of these interaction, for the HLA-C*07:02-RL9 complex, they had similar impact on the binding of both KIR2DL2 and 2DL3. Whether or not this is a general phenomenon or a particular feature of the HLA-C*07:02-RL9 remains to be determined. Nevertheless,*

the data indicate that the affinity of KIR2DL2 for its ligands is not strictly of higher affinity than that of KIR2DL3.”

“Figure [3]: is confusing as the asterisks do not apply to pairs of columns, but to comparisons between panels A and B.”

The figure has been modified to place KIR2DL2 and 2DL3 mutants on the same panel.

Reviewer #2 (Remarks to the Author):

Moradi et al. determined the crystal structures of HLA-Cw*07:02 complexed either with natural killer cell receptor KIR2DL2 or KIR2DL3, both of which have been believed to be specific for HLA-C1 group. They revealed similar but slightly different binding modes of these KIRs toward the same HLA-Cw*07:02. They further performed the mutagenesis study on KIRs expressed on 293 cells to identify some amino acids responsible for HLA binding and specificity. On the other hand, they also analyzed the binding specificity of KIR2DL2 and KIR2D3 tetramers toward the beads attached with 18 HLA allele proteins and cells. KIR2DL3 showed more specific binding to HLA-C1 group than KIR2DL2. Furthermore, they evaluated the activation of subsets of NK cells divided by expression levels of specific KIRs, again showing that KIR2DL3-expressing NK cells was more inhibited by HLA-C1 group than KIR2DL2-expressing ones. These results provide insights on the functional and structural differences of HLA-C

specific KIR2Ds. This paper is interesting and has some potential to advance our knowledge for a field of NK cell immunology, but they should discuss more about related reports such as Moesta, A.K. et al. *J Immunol* 180, 3969-79 (2008). I have some comments.

Comments

1. The authors should discuss the relationship of crystal structures of KIR2DL-HLA-C complexes with functional data of KIR mutants as well as HLA alleles, especially shown in Figures 5 and 6. Is it possible to explain these functional differences of HLA allele specificities from current structures?
2. Is there any clear correlation of functional data between Figures 5 and 6F? HLA-C1203 bound to KIR2DL3 only, while HLA-C1601 and C0304 did to both KIRs. However, the NK cell activation might not seem to be related to binding results.
3. In Figure 4, raw FCM data, at least representative data, should be presented. Did they check the expression levels of KIR mutants? How did they calculate % positive cells (more than 100% of positive cells?)?

Minor comments

1. The electron density map of some area such as binding interfaces should be shown.

2. In Figure 3, the authors tried to show the equilibrium binding experiments. However, KIR2DL2-Cw0304 binding show decreasing responses at saturated states, and KIR2DL3-Cw3 and –Cw7 both increasing responses. These seem to have either inappropriate control or nonspecific binding. How did they determine the responses for fitting? In principle, they should perform the binding with short time of saturation states to reduce the effect of nonspecific binding.

3. In Supplementary Figure 1, Asp187 of both KIRs seem to be located distantly from the F-F' loops.

4. In Figure 2D, the differences of interactions of binding amino acids are hardly understood, maybe because coloring and color depth are similar.

Response to reviewer #2

We thank the referee for their careful review of our paper and for stating ‘**this paper is interesting and has some potential to advance our knowledge for a field of NK cell immunology**’. The referee made a number of comments, which we address in turn below, that has resulted in improvements to our initial submission.

“1. The authors should discuss the relationship of crystal structures of KIR2DL-HLA-C complexes with functional data of KIR mutants as well as HLA alleles, especially shown in Figures 5 and 6. Is it possible to explain these functional differences of HLA allele specificities from current structures?”

We consider that the structures do provide important insights into functional differences. Namely, KIR2DL2 and KIR2DL3 bind the same peptide-HLA complex differently. From this structural observation, we draw the conclusion that the plasticity of KIR2DL binding to HLA-C allomorphs is attributable to the interdomain “twist” angle driven by residues 16 and 148. In the discussion we now state (page 11, line 29):

“This unexpected capacity to use identical residues to make alternate pHLA contacts appears to be driven by differences in the juxtapositioning and specifically the “twist” of the D1 and D2 domains” and in the results “this suggests that the polymorphisms at positions 16 and 148 are likely drivers of the relative positioning of the D1-D2 domains.”

And (page13, line 7):

*While the binding of the recombinant KIR2DL2/3 to HLA-coated beads exhibited the expected specificity for C1 allotypes, the extent to which modest differences between allotypes would be manifest in more physiological recognition settings is still unclear. For example, consistent with previous work, both binding and in vitro assays of cellular recognition indicate the HLA-C*07:02 is a strong ligand for both KIR2DL2 and KIR2DL3, yet exactly where it sits in the hierarchy of C1 allotypes is likely impacted by numerous factors. For example, both the nature of the peptide repertoire and density of each HLA-C allotype present on the beads may significantly impact the extent of KIR tetramer binding. Indeed, the processes of purifying each HLA allotype and then generating stably coated beads might ultimately bias the bead-associated HLA-C to display a repertoire of peptides that has exceptional capacity to stabilise the HLA relative to that present on the cell surface. Similarly, the cellular analyses are also likely acutely dependent on both the peptide repertoire present on the surface of the target cell as well as the overall level of HLA-C expression. Indeed, overexpression of the HLA-C allotypes as is likely in the transfected 721.221 cells may well act to compress differences in recognition by KIR2DL2/3. Similarly, in the functional analyses NK cell intrinsic factors such as differences in the KIR2DL2/3 allotypes expressed, the co-expression of activating KIR2DS, and education likely also contribute to functional outcomes, albeit that all the donors used in the analyses here possessed C1 alleles and similar responses were observed to 221 cells by not only KIR2DL2⁺/S2⁻ and KIR2DL3⁺ cells as well as those the expressed KIR2DS2.*

“Is there any clear correlation of functional data between Figures 5..? HLA-C1203 bound to KIR2DL3 only, while HLA-C1601 and C0304 did to both KIRs. However, the NK cell activation might not seem to be related to binding results.”

There is no clear correlation between the binding data and the functional analyses. Our structural, binding and functional analyses all demonstrate that C1 allotypes differ in their capacity to interact with KIR2DL2 and KIR2DL3. However while Fig 5 is a binding assay, measuring the strength of the interaction between distinct HLA-C allotypes and defined KIR2DL2/3 allotypes, the capacity of to inhibit NK cell activation is dependent on: (a) the extent of education, (b) the level of expression of each HLA-C allotype on transfected .221 cells, (c) the alleles of KIR2DL2/3 expressed on the NK cells used in Fig 6 and (d) the strength of the interaction between KIR2DL2/3 and each HLA-C allotype. There are however clear differences in both binding and functional recognition between C1 allotypes and the C2

allotype HLA-C*04:01 which neither inhibits functional responses by KIR2DL2/3+ NK cells nor appreciably binds KIR tetramer. The lack of a strict correlation is now addressed in the discussion as below (page 13, line 5):

*However, the hierarchy of HLA-C allotype recognition observed in our functional analyses did not strictly correlate with that from the direct binding analyses. While the binding of the recombinant KIR2DL2/3 to HLA-coated beads exhibited the expected specificity for C1 allotypes, the extent to which modest differences between allotypes would be manifest in more physiological recognition settings is still unclear. For example, consistent with previous work, both binding and in vitro assays of cellular recognition indicate the HLA-C*07:02 is a strong ligand for both KIR2DL2 and KIR2DL3, yet exactly where it sits in the hierarchy of C1 allotypes is likely impacted by numerous factors. For example, both the nature of the peptide repertoire and density of each HLA-C allotype present on the beads may significantly impact the extent of KIR tetramer binding. Indeed, the processes of purifying each HLA allotype and then generating stably coated beads might ultimately bias the bead-associated HLA-C to display a repertoire of peptides that has exceptional capacity to stabilise the HLA relative to that present on the cell surface. Similarly, the cellular analyses are also likely acutely dependent on both the peptide repertoire present on the surface of the target cell as well as the overall level of HLA-C expression. Indeed, overexpression of the HLA-C allotypes as is likely in the transfected 721.221 cells may well act to compress differences in recognition by KIR2DL2/3. Similarly, in the functional analyses NK cell intrinsic factors such as differences in the KIR2DL2/3 allotypes expressed, the co-expression of activating KIR2DS, and education likely also contribute to functional outcomes, albeit that all the donors used in the analyses here possessed C1 alleles and similar responses were observed to 221 cells by not only KIR2DL2⁺/S2⁻ and KIR2DL3⁺ cells as well as those that expressed KIR2DS2.*

“In Figure 4 [now 3], raw FCM data, at least representative data, should be presented. How did they calculate % positive cells (more than 100% positive cells)?

To clarify, the % positive cell being more than 100% is a matter of normalisation. The cells that were transfected with the KIR were stained with anti-FLAG antibody to ID the percentage of cells transfected. The cells were also separately stained with HLA tetramers and that percentage was recorded. The % tetramer positive cells = (%tet+)/(%FLAG+) *100. The figure has now been updated showing representative patterns of FLAG and KIR specific mAb along with tetramer staining of transfected cells

expressing the wild type receptors. We have also clarified how the data were generated in the figure legend.

Minor comments.

“Electron density map ...should be shown”

Representative electron density maps for the KIR2DL2 and 2DL3 HLA-Cw7-RL9 complexes have been added as Supplementary Figure 1.

“In Figure 3, the authors tried to show equilibrium binding experiments ... they should perform the binding ... to reduce the effect of non-specific binding”

The SPR experiments were redone on a BIAcore T3000 with a CM5 chip. A high flow rate was used and a W6/32 antibody alone flow cell used as the control for data subtraction. This reduced the effects of non-specific binding. This data is shown as Supplementary Figure 6.

“In Supp Figure 1, Asp 187 ... seem to be located distantly...”

The figure has been fixed to show Asp187 connected to the ribbon.

“In Figure 2D, the differences of interactions of binding amino acids are hardly understood, maybe because coloring and color depth are similar”

The colour of these labels has been changed to aid in visualization.

Reviewer #3 (Remarks to the Author):

Understanding the molecular basis for genetic associations of KIR and their HLA class I ligands with disease is a key question in the field. Multiple associations have demonstrated a difference in disease risk between individuals who carry KIR2DL2 and KIR2DL3, two highly related receptors that bind the same ligand, HLA-C allotypes of group 1 (C1).

The manuscript by Moradi et al. describes novel crystal structures of KIR2DL2 and KIR2DL3 bound to the same HLA-C*07:02 ligand, allowing a direct comparison of ligand recognition. These structures and the accompanying tetramer binding studies that map the specific binding sites are novel contributions. However, the manuscript provides little on the question of why disease risk differs between individuals who carry KIR2DL2 and KIR2DL3.

The major conclusion that KIR2DL2 is a weaker receptor than KIR2DL3 and that this may be due to the differing KIR docking geometries is not consistent with the data presented. Notably, in the bead binding assay KIR2DL2 is a stronger receptor than KIR2DL3 for C*07:02, while by surface plasmon resonance they are the same. Thus, KIR2DL2 is not a weaker receptor than KIR2DL3 for C*07:02. The different docking geometries must be incidental to differences in binding strength.

The authors allude to the role of peptide sequence in modifying KIR recognition of HLA-C but do not cite key references on this topic. Binding of KIR2DL2 and KIR2DL3 to C1 allotypes is quite sensitive to the peptide sequence. A limitation of their structural analysis is that it is based on C*07:02 in the context of only one peptide sequence.

NK cell degranulation assays showed stronger inhibition by KIR2DL3+ compared to KIR2DL2+ NK cells, supporting their conclusion that KIR2DL2 is a weaker receptor than KIR2DL3. However, the interpretation of these experiments is significantly impacted by the inability to eliminate the contribution of the activating receptor KIR2DS2, since the antibody used does not discriminate between KIR2DL2 and KIR2DS2. NK cells that coexpress these two receptors may have diminished inhibition due to the contribution of KIR2DS2. Although the authors note this shortcoming, they do not consider it when drawing their conclusion.

Multiple publications have noted that KIR2DL3 binding to C1 allotypes is weaker than KIR2DL2 on cells or beads when using soluble KIR-Fc fusion proteins. The reason(s) for this weaker binding is still not understood. The authors used refolded KIR tetramers for their bead binding experiments and observed that KIR2DL2 binds more weakly to C1 allotypes than KIR2DL3. However, the authors did not resolve this discrepancy.

Response to reviewer #3

We thank the reviewer for their critique of our initial submission, and for stating that the structural work and tetramer binding studies represent ‘**novel contributions**’. They made a number of valuable comments, which we address in turn below.

“The manuscript offers little on the question of why disease risk differs...”

The differences between KIR2DL2 and KIR2DL3 on disease progression have been noted at the genetic association level (Khakoo et al Science 2004 and Romero et al Mol Immunol 2008). These studies have suggested the “weaker” inhibitory potential of KIR2DL3 provides a more robust immune response to HCV challenge. Our data suggest that KIR2DL3 is not necessarily a “weaker” receptor than KIR2DL2. Further, we show that KIR2DL3 and 2DL2 have different hierarchies of binding affinity for HLA-C alleles, and again 2DL3 is not necessarily the weaker binder. Thus, our results impact on the interpretation of the genetic association data. However, our data do not provide a “why” or mechanistic basis for disease risk. So, we have tempered our conclusion and have altered the concluding remarks of the discussion to now read:

“Thus, studies that further refine the impact of polymorphism in both receptors and ligands and that also account for the frequency of cells expressing both these and genetically linked receptors such as KIR2DS2 will be needed to better understand how these innate receptor/ligand combinations contribute to the control of infectious disease and cancer.”

“The major conclusion that KIR2DL2 is a weaker receptor than 2DL3 and that this may be due to the differing KIR docking geometries is not consistent with the data presented”

It was not our intention to state this in such stark terms. We have done further SPR analyses of the interaction between KIR2DL2 and KIR2DL3 and multiple HLA-C*07:02-peptide complexes (Supplementary Fig 6), showing that in each case the affinities for KIR2DL2 and 2DL3 binding are very similar. Our Luminex analyses across a broader array of C1 allotypes suggests that KIR2DL2 does bind better than KIR2DL3 to some but not all allotypes. This is now reflected in the following comment (page 9, line 4):

*“individual HLA-C allomorphs also showed differences in binding preference between the two KIR2DL receptors. For example, HLA-C*08:01 and HLA-C*12:03, bound considerably better to KIR2DL3 than to KIR2DL2, whilst HLA-C*07:02 was bound with higher avidity by KIR2DL2”.*

Critically, having performed new functional analyses on the KIR2DL2+/KIR2DS2- subset (Figure 6 and Supplementary Fig 8), which showed the inhibitory potential response of the KIR2DL2/S2⁺ was in effect diluted by the presence of cells expressing KIR2DS2, we have now revised our conclusions stating (page 10, line 34):

“that the KIR2DL2⁺S2⁻ subset had a similar pattern of C1 allotype recognition to the KIR2DL3⁺ subset against each HLA-C allotype tested.”

“Notably, in the bead binding assay KIR2DL2 is a stronger receptor than KIR2DL3 for C*07:02, while by SPR they are the same”.

This is correct. The bead binding assay is performed with HLA presenting a repertoire of cellular-derived peptides whilst the SPR is performed with a single peptide. Thus, KIR2DL2 appears to be more tolerant to cellular peptide repertoire in HLA-C*07:02 than 2DL3. The RL9 peptide was chosen for SPR analysis as this was the peptide from which the structures were generated. It is a very high affinity peptide and so is likely not reflective of the majority of peptides on the cell surface. We broadened the SPR analysis to include 14 P7 and P8 mutants of the RL9 peptide. These did not show significant differences between KIR2DL2 and KIR2DL3 binding, likely because of the high affinity of the RL9 peptide (Supplementary Figure 6). We now comment on this in the discussion as below

*“Further, while substitutions at positions 7 and 8 clearly impacted the affinity of these interaction, for the HLA-C*07:02-RL9 complex, they had similar impact on the binding of both KIR2DL2 and 2DL3. Whether or not this is a general phenomenon or a particular feature of the HLA-C*07:02-RL9 remains to be determined.”*

and more broadly (page 12, line 26):

Thus while KIR2DL2 may bind some HLA-C/peptide combinations better than KIR2DL3, our data suggests that are also HLA-C/peptide combinations that will be preferentially recognised by KIR2DL3. Moreover, given the potential of peptide to modulate recognition, whether or not a particular HLA-C allotype is preferentially bound by KIR2DL2 or KIR2DL3 is likely dependent on the cellular peptide repertoire and by implication the type of cell and potentially the extent to which its repertoire is modified by transformation and/or infection.

“Thus, KIR2DL2 is not a weaker receptor than KIR2DL3 for C*07:02. The different docking geometries must be incidental to differences in binding strength”.

We agree that it is very difficult to relate binding strength with studying intermolecular contacts derived from structural data. However, we do observe very clear differences in binding geometries, and this is coincident with clear differences in binding hierarchies to HLA-C allomorphs as observed in the bead binding assay which incorporates a repertoire of cellular-derived peptides. In the substantially revised manuscript, we have tempered our conclusions to reflect the concern of the reviewer.

“The authors allude to the role of the peptide sequence in modifying KIR recognition ... but do not cite key references...”

We have expanded the references to peptide recognition in the discussion (page 12, line 15).

“...contrasted with previous studies using KIR2DL-Ig fusion proteins which suggested that KIR2DL2 was a stronger and more peptide tolerant receptor than KIR2DL3^{12,36-38}”.

“Binding of KIR2DL2 and 2DL3 to C1 allotypes is sensitive to peptide sequence. A limitation of their structural analysis is that it is based on C*07:02 in the context of a single peptide..”

Indeed, structural studies are limited to single peptides. However, this also a strength of our study as we can conclude from the use of a single peptide that the structural differences we observe between KIR2DL2 and 2DL3 bound to HLA-C*07:02 are independent of the peptide sequence.

“NK cell degranulation assays showed stronger inhibition by KIR2DL3+ compared to KIR2DL2+ NK cells, supporting their conclusion that KIR2DL2 is a weaker receptor than KIR2DL3. However, the interpretation of these experiments is significantly impacted by the inability to eliminate the contribution of the activating receptor KIR2DS2, since the antibody used does not discriminate between KIR2DL2 and KIR2DS2. NK cells that co-express these two receptors may have diminished inhibition due to the contribution of KIR2DS2. Although the authors note this shortcoming, they do not consider it when drawing their conclusion.”

We have now addressed the contribution of KIR2DS2 more formally through the use of the 1F12 mAb which allows us to identify cells that are KIR2DL2+/KIR2DS2- and thus compare them with those that express KIR2DL3 (see above). The diminished levels of inhibition seen in the GL183+/KIR2DL3- fraction were associated with cells that express KIR2DS2. The abstract, results and discussion have now

been amended to reflect our observations that there are at best only modest differences between the KIR2DL2+ KIR2DS2- subset and cells that express KIR2DL3.

“Multiple publications have noted that KIR2DL3 binding to C1 allotypes is weaker than KIR2DL2 on cells or beads when using soluble KIR-Fc fusions proteins... the authors have used refolded KIR tetramers... and observed that KIR2DL2 binds more weakly to C1 allotypes than KIR2DL3. However, the authors do not resolve this discrepancy.”

We thank the reviewer for this insight. We suggest that the differences observed between Ig-fusion KIR (dimeric and from mammalian cell expression) and streptavidin-fusion KIR (tetrameric and from *E. coli* expression) may be due to the glycosylation and the oligomerisation state of the complexes. We have updated the discussion to reflect this point (page 12, line 15):

“This contrasted with previous studies using KIR2DL-Ig fusion proteins which suggested that KIR2DL2 was a stronger and more peptide tolerant receptor than KIR2DL3^{12,36-38}. There are significant intrinsic differences between the refolded KIR2DL2/3 compared with dimeric-KIR-Ig fusion proteins that include differences in glycosylation and the oligomerisation of the receptor which may impact docking geometries and/or modify the relative positioning of the D1 and D2 domains when bound to HLA-C.”

We would also note that our SPR analyses which did not involve multimeric recombinant proteins also did not show that KIR2DL2 bound with higher affinity to a range of ligands than KIR2DL3.

Reviewers' comments:

Reviewer #1 (Remarks to the Author):

The authors have answered the comments satisfactorily. However one major concern is that in their new supplementary Figure 6 it appears that KIR2DL2 bind to P7G, P7A and P8G with higher affinity than KIR2DL3 (and also possibly p7N, although SD is quite large). This is important as it may impact the conclusion of the paper that there is no peptide selectivity differences between the receptors. If this were a random effect within the assay then it would be expected that KIR2DL3 would bind at least one of the peptides with a higher affinity, but this does not appear to be the case. At the least these data need to be moved to the main figure, a statistical comparison made and the data need to be discussed in detail.

Reviewer #2 (Remarks to the Author):

The authors have addressed adequately the concerns raised in my review.

Reviewer #3 (Remarks to the Author):

The authors deserve to be commended for the major effort undertaken during revision of this interesting study. A number of new experiments were performed to address the major comments of the reviewers, which happened to be in agreement for a majority of their comments. A new antibody was used that allowed a distinction between NK cells expressing KIR2DL2 either in the absence or presence of KIR2DS2. The KIR2DL2+S2⁻ cells could then be compared with KIR2DL3 cells. A panel of mutations were made in KIR2DL2 and KIR2DL3 and each one of them produced as tetramers to study binding to two C1 allotypes of HLA-C. All of these experiments confirmed that KIR2DL2 and KIR2DL3 are rather similar in their reactivity to HLA-C allotypes. They reinforce the overriding conclusion that KIR2DL2 and KIR2DL3 differ mainly by the orientation and angles of the D1 and D2 domains and the molecular contacts with HLA-C. Strikingly, the topology of the KIR2DL3 interaction with HLA-C*07:02 is very similar to that of KIR2DL1 interaction with the C2 allotype HLA-C*04:01 while it is quite different from the KIR2DL2 interaction with the same C1 allotype HLA-C*07:02. Furthermore, KIR2DL2 compared to itself has different orientation and contacts with HLA-C*07:02 and another C1 allotype, HLA-C*03:04. This suggests that the inhibitory KIR2DL receptors have the ability to adapt to the many different HLA-C allotypes they encounter and that this adaptation may be constrained by the peptides bound to HLA-C. It is likely that KIR2DL3 also has the flexibility to adapt to different HLA-C allotypes. No further revision is required. The authors may choose to discuss some of the points raised here if they wish.

Reviewer #1 (Remarks to the Author):

The authors have answered the comments satisfactorily. However one major concern is that in their new supplementary Figure 6 it appears that KIR2DL2 bind to P7G, P7A and P8G with higher affinity than KIR2DL3 (and also possibly p7N, although SD is quite large). This is important as it may impact the conclusion of the paper that there is no peptide selectivity differences between the receptors. If this were a random effect within the assay then it would be expected that KIR2DL3 would bind at least one of the peptides with a higher affinity, but this does not appear to be the case. At the least these data need to be moved to the main figure, a statistical comparison made and the data need to be discussed in detail.

Reviewer #2 (Remarks to the Author):

The authors have addressed adequately the concerns raised in my review.

Reviewer #3 (Remarks to the Author):

The authors deserve to be commended for the major effort undertaken during revision of this interesting study. A number of new experiments were performed to address the major comments of the reviewers, which happened to be in agreement for a majority of their comments. A new antibody was used that allowed a distinction between NK cells expressing KIR2DL2 either in the absence or presence of KIR2DS2. The KIR2DL2+S2- cells could then be compared with KIR2DL3 cells. A panel of mutations were made in KIR2DL2 and KIR2DL3 and each one of them produced as tetramers to study binding to two C1 allotypes of HLA-C. All of these experiments confirmed that KIR2DL2 and KIR2DL3 are rather similar in their reactivity to HLA-C allotypes. They reinforce the overriding conclusion that KIR2DL2 and KIR2DL3 differ mainly by the orientation and angles of the D1 and D2 domains and the molecular contacts with HLA-C. Strikingly, the topology of the KIR2DL3 interaction with HLA-C*07:02 is very similar to that of KIR2DL1 interaction with the C2 allotype HLA-C*04:01 while it is quite different from the KIR2DL2 interaction with the same C1 allotype HLA-C*07:02. Furthermore, KIR2DL2 compared to itself has different orientation and contacts with HLA-C*07:02 and another C1 allotype, HLA-C*03:04. This suggests that the inhibitory KIR2DL receptors have the ability to adapt to the many different HLA-C allotypes they encounter and that this adaptation may be constrained by the peptides bound to HLA-C. It is likely that KIR2DL3 also has the flexibility to adapt to different HLA-C allotypes. No further revision is required. The authors may choose to discuss some of the points raised here if they wish.

Reviewer #1

Query 1: ***“The authors have answered the comments satisfactorily. However one major concern is that in their new supplementary Figure 6 it appears that KIR2DL2 bind to P7G, P7A and P8G with higher affinity than KIR2DL3 (and also possibly p7N, although SD is quite large). This is important as it may impact the conclusion of the paper that there is no peptide selectivity***

differences between the receptors. If this were a random effect within the assay then it would be expected that KIR2DL3 would bind at least one of the peptides with a higher affinity, but this does not appear to be the case. At the least these data need to be moved to the main figure, a statistical comparison made and the data need to be discussed in detail.”

We thank the reviewer for stating that we have addressed their comments satisfactorily. They raise one concern about Supplementary Figure 6, which we address below.

As requested, we have now performed statistical analysis on the SPR results and reported the *p* and *q* values from multiple unpaired t-tests. These results have now been moved into the main text as Table 1. The Supplementary Figure 6 remains a display of the data acquisition. Although P7G, P7A and P8G appear to have marginally higher affinities to KIR2DL3 the differences were not statistically significant. It is our conclusion that we cannot state that KIR2DL2 and KIR2DL3 differ in their affinity for the tested P7 and P8 mutants of the RL9 peptide. We state in the results “*Thus, at least in the context of the HLA-C*07:02-RL9 peptide, there were no marked differences between KIR2DL2 and 2DL3 with respect to their sensitivity to P7 and P8 substitutions.*”

The reviewer has stated that this finding “***.. is important as it may impact the conclusion of the paper that there is no peptide selectivity differences between the receptors.***”

The data above does not impact on the conclusions of the paper. To clarify, we do observe peptide selectivity between KIR2DL2 and KIR2DL3 in the bead-binding assay which incorporates an unknown yet presumably large number of cellular-derived peptides. We have clarified this point in the discussion:

*“KIR2DL2 bound select allotypes such as HLA-C*15:02 and -C*17:01 better than KIR2DL3, which in turn had stronger interactions with HLA-C*08:01 and -C*12:03 than KIR2DL2. Critically, given the potential of peptide variation to impact KIR binding, these broader specificity analyses of KIR2DL2 and -2DL3 used bead bound HLA-C complexed with a cellular-derived peptide repertoire rather than a single peptide.”*

As the SPR experiments are dependent on single peptides built on the RL9 core peptide sequence there is a risk of over-interpreting the SPR results and applying it broadly across all HLA-peptide complexes. We suggest that the SPR analysis be viewed within the context of HLA-C*07:02 and the RL9-peptide and so state in the discussion:

*“...while substitutions at positions 7 and 8 clearly impacted the affinity of these interactions, for the HLA-C*07:02-RL9 complex, they had similar impact on the binding of both KIR2DL2 and 2DL3. Whether or not this is a general phenomenon or a particular feature of the HLA-C*07:02-RL9 remains to be determined.”*

Reviewer #2

We thank reviewer #2 for their positive evaluation of our resubmitted paper. No comments to address.

Reviewer #3

We thank reviewer #3 for their positive evaluation of our resubmitted paper. No comments to address.